# Retrieval of contextual memory can be predicted by CA3 remapping and is differentially influenced by NMDAR activity in rat hippocampus subregions

**Magdalena Miranda**[1,¤a], **Azul Silva**[2,3], **Juan Facundo Morici**[1,¤b], **Marcos Antonio Coletti**[2,3], **Mariano Belluscio**[2,3,‡]*, **Pedro Bekinschtein**[1,‡]*

**1** Laboratorio de Memoria y Cognición Molecular, Instituto de Neurociencia Cognitiva y Traslacional, CONICET-Fundación INECO-Universidad Favaloro, Ciudad Autónoma de Buenos Aires, Buenos Aires, Argentina, **2** Laboratorio Bases neuronales del comportamiento, Departamento de Ciencias Fisiológicas, Facultad de Ciencias Médicas, Universidad de Buenos Aires, Ciudad Autónoma de Buenos Aires, Buenos Aires, Argentina, **3** CONICET—Universidad de Buenos Aires, Instituto de Fisiología y Biofísica Bernardo Houssay (IFIBIO Houssay), Ciudad Autónoma de Buenos Aires, Buenos Aires, Argentina

¤a Current address: Institute of Functional Genomics, University of Montpellier, CNRS, INSERM, Montpellier, France
¤b Current address: Institut du Fer à Moulin, INSERM and Sorbonne Université, Paris, France
☯ These authors contributed equally to this work.
‡ MB and PB also contributed equally to this work.
* mbellu@fmed.uba.ar (MB); pbekinschtein@favaloro.edu.ar (PB)

**Data Availability Statement:** The data that support these findings is available in OSF at https://osf.io/7pw23/.

## Abstract

Episodic memory is essential to navigate in a changing environment by recalling past events, creating new memories, and updating stored information from experience. Although the mechanisms for acquisition and consolidation have been profoundly studied, much less is known about memory retrieval. Hippocampal spatial representations are key for retrieval of contextually guided episodic memories. Indeed, hippocampal place cells exhibit stable location-specific activity which is thought to support contextual memory, but can also undergo remapping in response to environmental changes. It is unclear if remapping is directly related to the expression of different episodic memories. Here, using an incidental memory recognition task in rats, we showed that retrieval of a contextually guided memory is reflected by the levels of CA3 remapping, demonstrating a clear link between external cues, hippocampal remapping, and episodic memory retrieval that guides behavior. Furthermore, we describe NMDARs as key players in regulating the balance between retrieval and memory differentiation processes by controlling the reactivation of specific memory traces. While an increase in CA3 NMDAR activity boosts memory retrieval, dentate gyrus NMDAR activity enhances memory differentiation. Our results contribute to understanding how the hippocampal circuit sustains a flexible balance between memory formation and retrieval depending on the environmental cues and the internal representations of the individual. They also provide new insights into the molecular mechanisms underlying the contributions of hippocampal subregions to generate this balance.

**Funding:** This work was supported by the Consejo Nacional de Investigaciones Científicas y Técnicas (PUE 0052; https://www.conicet.gov.ar) to PB, the FONCYT (PICT 2018-1062; https://www.argentina. gob.ar/ciencia/agencia) to PB, the FONCYT (PICT 2021-4663; https://www.argentina.gob.ar/ciencia/ agencia) to MB and the Consejo Nacional de Investigaciones Científicas y Técnicas (PIP 2656; https://www.conicet.gov.ar) to MB. The funders had no role in study design, data collection and analysis, decision to publish, or preparation of the manuscript.

**Competing interests:** The authors have declared that no competing interests exist.

**Abbreviations:** AC, all cue; AIC, Akaike information criterion; GLM, general linear model; HP, hippocampus; LRT, likelihood ratio test; L-VGCC, L-type voltage gated calcium channel; NC, no cue; PBS, phosphate-buffered saline; PC, partial cue; RM, repeated measures; t-SNE, t-distributed stochastic neighbor embedding.

## Introduction

Learned behavior is the outcome of an interaction between the environment and representations stored in memory. Many contextual responses depend on whether experiences are novel or similar to previous ones. Since the environment is continuously changing, episodic memory retrieval usually occurs under contextually modified conditions and the ability to retrieve previous memories despite partial contextual change becomes crucial [1]. Changes in environmental cues force the brain to make a difficult choice: should we discard differences between 2 similar experiences (i.e., retrieve an already stored representation) or should we build a new memory? Under each particular context, the brain evaluates environmental and internal cues to determine whether a context is familiar or not. Based on the proposed dual role of the hippocampus (HP) in context recognition, computational studies suggested the need for 2 independent systems [2] that optimize information processing under small or large contextual changes. These studies proposed that unique features of hippocampal subregions could allow the development of computationally distinct and complementary functions needed for correct episodic memory formation and retrieval. These processes, based on the activity of the DG-CA3 circuit, are known as pattern separation and pattern completion [2–4].

Hippocampal cells can represent space (place cells) [5], time [6,7], events [8–10], or a conjunctive representation of them [11–13]. The HP can represent different contexts as independent maps [14–16]. The process that leads to a reorganization of cell ensemble activity when the context changes are known as remapping. Depending on the conditions of spatial shifts, remapping can be complete or partial [17].

But can remapping explain retrieval or encoding of spatial memories that sustain behavior? Despite the intuition of place cell importance for episodic memory, there is still no clear evidence that remapping is directly related to the expression of different contextual memories. One reason for this is that most studies have only related remapping to observable properties of the environment but not to behavioral outputs that could give some insight into the inferences the animal is making about a particular environment or experience (i.e., its internal context representation). In other words, the variability of circuit activation in the same physical environment has been overlooked, variability that could be influenced by motivation [18,19], attention [20], and experience [21]. Moreover, it is not clear how individual mnemonic variability (i.e., behaving as being in a familiar context or not) relates to internal context representation and place cell activity.

Experimental evidence and computational models suggest a role of the DG-CA3 in pattern separation and pattern completion, both at the electrophysiological and behavioral level [22–24]. In this regard, previous studies have implicated NMDA receptors (NMDARs) in DG and CA3 in behavioral memory discrimination and rate remapping [25,26]. In addition, it has been shown that NMDARs are particularly relevant to recover memory from a cue-degraded context [26] and to allow CA1 neurons to maintain place field characteristics between these conditions [27,28]. Since some theories suggested that the DG-CA3 circuit of the HP mediates the dynamic competition between memory differentiation (supported by pattern separation) and generalization (supported by pattern completion), understanding the nature of this interaction could provide a framework to explain how episodic memory is rooted on contextual representation.

In this work, we study the interaction between pattern separation and pattern completion during object in context memory at the behavioral (i.e., memory retrieval), molecular, and electrophysiological (i.e., remapping) level. To model episodic memory retrieval in an incidental task without the bias of motivational influences over contextual representations, we developed a context-degraded task. Taking into account the fundamental role of the balance

between discrimination and generalization for episodic memory function, we decided to test the role of molecular mechanisms as potential modulators of this balance in the DG-CA3 circuitry and analyze CA3 neuronal activity. We found that internal context representation, seen as the efficacy of memory retrieval to guide behavior, could be explained by CA3 neuronal context representation. Furthermore, we showed that the limits between retrieval of a previous experience and encoding of a new one can be shifted to favor one process or the other by modulating NMDARs activity in the DG-CA3 circuitry.

## Results

### The associative retrieval task as a measure of the ability to retrieve original experiences under a cue-degraded condition

Rodent's inherent preference for novelty requires engaging retrieval to determine an event as familiar or novel. To determine whether rats would be able to retrieve experiences under degraded environmental cues, we adapted the spontaneous object recognition task to an associative retrieval task [29–31]. During a training session, animals incidentally associated a novel object with a context with 6 distal cues. They were later tested with an identical copy of the object in the same location but under a variable number of the original contextual cues: "all cues" (AC), "partial cues" (PC, 3 distal cues), and "no cues" (NC, no distal cues) conditions (Fig 1A, 1B and 1E; see SI Appendix). For pharmacological experiments, animals were tested 24 h after the training session (2-day version) and for the electrophysiological experiments, animals were tested 4 h after the training session (1-day version). The time animals spent exploring the object during the test session was considered a measure of object memory in that context, with decreased object exploration when animals retrieve the original object-context memory. We found no difference in object exploration time between groups during training (Fig 1C and 1F). Since all groups were trained in AC conditions, this indicates all groups were, a priori, equal in terms of object exploration levels. Critically, there was a significant condition effect over the percentages of object exploration during test. Animals in the NC condition had significantly higher percentages of object exploration than the AC and PC conditions (Fig 1D and 1G). Percentages of object exploration were significantly lower than 100% for both the AC and PC, but not for the NC in both versions of the task. The absence of differences due to the cue removal in the PC condition suggests that rats are able to retrieve the original representation using degraded information. Rearing behavior, an alternative measure of environmental novelty less dependent on object memory than object exploration time [32,33], showed a similar tendency (S1 Fig).

### Contextual cues in the task are important to guide reactivation of the object memory in the Prh

Under certain conditions, memory reactivation can increase susceptibility to post-retrieval protein synthesis inhibitors, specifically targeting the reactivated memory trace. Inactive memories during the test remain stable and are unaffected by protein synthesis inhibition [34–39]. If contextual information is enough to guide retrieval of the original object-in-context memory in our task, exposure to the context alone should destabilize object memory in regions where this memory is stored, like the perirhinal cortex (Prh) [34,37,40]. To address this question, animals trained on day 1 in the task were exposed to a retrieval session on day 2 in the same context but in the absence of an object in either AC or NC conditions (TS1). Immediately after context exposure, they were infused into the Prh with the protein synthesis inhibitor Emetine (Eme) or with Vehicle (Veh). Object memory was evaluated 24 h later (day 3) on a test session

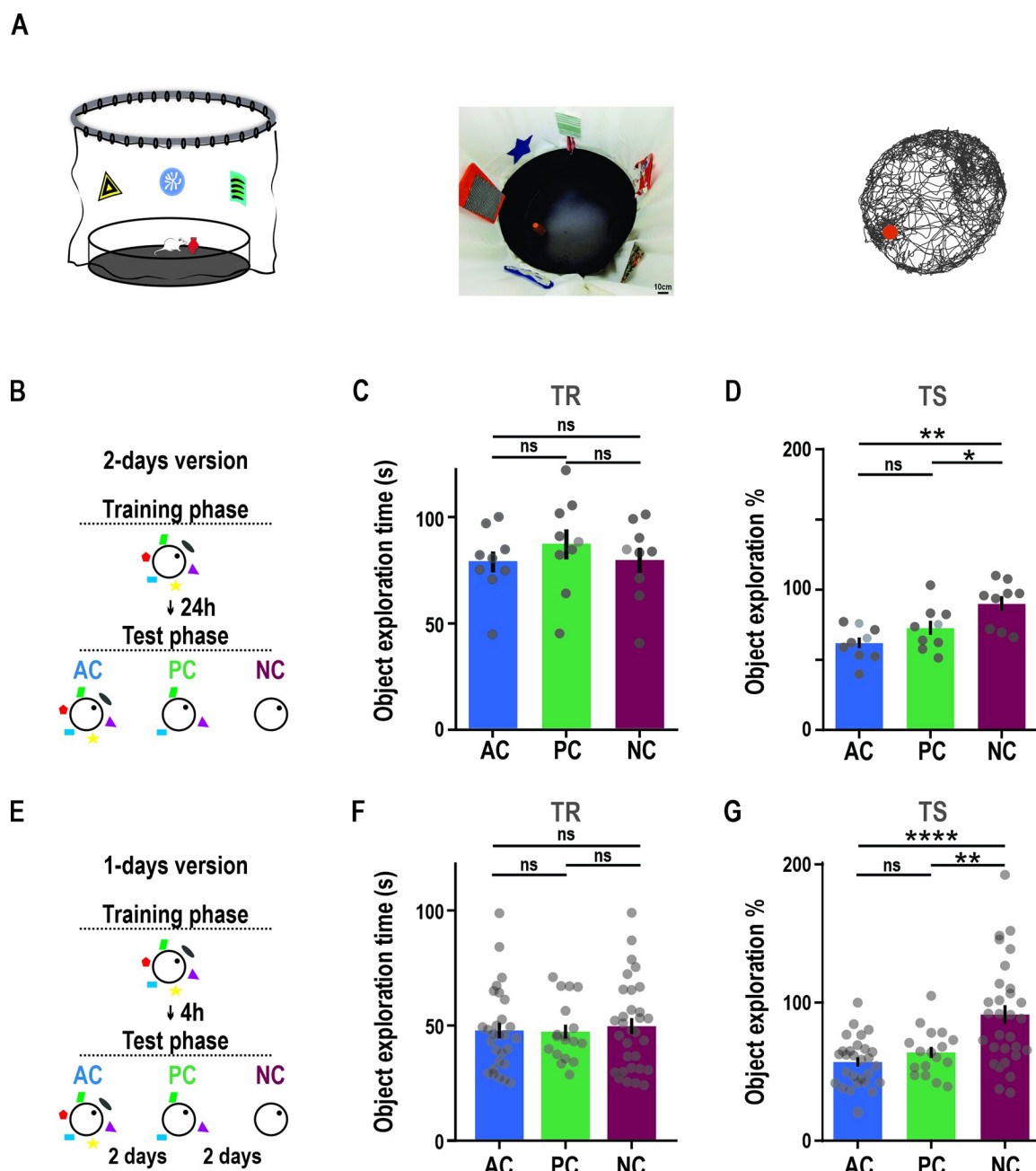

**Fig 1. The associative retrieval task. (A)** Associative retrieval task. "All cues" (AC, blue), "partial cues" (PC, green), and "no cues" (NC, orange) conditions. Right panel: representative animal trajectory. **(B)** Schematic illustration of 2-day version of the task. **(C)** Total object exploration time during training session for AC, PC, and NC during the 2-day version of the task. Rats spent an equal amount of time exploring the object during the training phase under the AC, PC, and NC conditions. One-way RM ANOVA F = 0.74, $p$ = 0.453, $n$ = 9. AC: 78.91 ± 5.38, PC: 87.23 ± 7.54, NC: 79.73 ± 6.31. **(D)** Percentage of object exploration in the presence of a variable number of cues (AC, NC, and PC) in the test session of the 2-day version of the task with respect to training. One-way RM ANOVA, F = 8.27, $p$ = 0.006; AC-NC $p$ = 0.008, PC-NC $p$ = 0.036; one sample $t$ test against 100% AC t = 9.39, $p$ < 0.0001; PC t = 5.18, $p$ = 0.0008; NC t = 1.83, $p$ = 0.105. AC: 61.84 ± 4.07, PC: 72.56 ± 5.29, NC: 89.91 ± 5.51. **(E)** Schematic illustration of the 1-day version of the task. **(F)** Total object exploration time during training session for AC, PC, and NC during the 1-day version of the task. Two-way RM ANOVA, interaction: F = 0.5755, $p$ = 0.7484, $n$ = 10, sessions = 75, each dot represents a session. AC: 47.89 ± 3.38, PC: 47.39 ± 3.07, NC: 49.74 ± 3.50. **(G)** Percentage of object exploration in presence of a variable number of cues (AC, NC, and PC) in the test session of the 1-day version of the task with respect to training. Two-way RM ANOVA, main effect condition F = 16.04, $p$ <0.0001. Tukey's post hoc test: $p$ < 0.0001 AC vs. NC; $p$ = 0.25 AC vs. PC; $p$ = 0.001 NC vs. PC. One sample $t$ test against 100% AC t = 14.45, $p$ < 0.0001; PC t = 8.76, $p$ < 0.0001; NC t = 0.37, $p$ = 0.91. AC: 56.98 ± 3.31, PC: 63.93 ± 4.11, NC: 92.13 ± 7.34. TR, training session; TS, test session. * $p$ < 0.05, ** $p$ < 0.01, **** $p$ < 0.0001. Individual values used to calculate mean and SEM are presented as dots. The data that support these findings is available in OSF at https://osf.io/7pw23/. AC, all cue; NC, no cue; PC, partial cue; RM, repeated measures.

(TS2), by placing the original object next to a novel one on a novel triangular context lacking any contextual cues so that object memory could be estimated independently of any contextual contribution (Fig 2A, see Methods section). Vehicle-infused animals had significantly higher discrimination ratios than zero during this last session, evidencing 48 h-object memory. On the other hand, Emetine infusion in the Prh significantly decreased the discrimination ratio in the "all cues" condition when compared to Vehicle, while no effect was seen in the "no cues" condition (Fig 2B) (one sample $t$ test against 0: NC-Veh $t = 4.77$, $p = 0.003$; NC-Eme $t = 4.22$, $p = 0.008$; AC-Veh $t = 4.38$, $p = 0.005$; AC-Eme $t = 1.30$, $p = 0.252$). These results indicate that the context associated with the object is sufficient to guide the reactivation of the original "object-in-context" memory.

To rule out nonspecific effects of Emetine, we tested the contextual dependency of memory labilization. During the training session, rats were exposed to an object in the NC condition and after 2 h to another object in the AC condition in a counterbalanced manner, and 24 h later, animals were placed in either the empty NC context or the empty AC context and both groups received Emetine or Vehicle infusion immediately after context exposure. Memory for the original objects was tested 24 h later against a novel object in both cases in a counterbalanced manner (Fig 2C, see Methods section). Time spent exploring each object during training did not differ between the AC and NC context (Fig 2D). Nevertheless, animals infused with Emetine after "no cues" exposure only showed a reduction of the discrimination ratio for the object originally associated with that condition. On the other side, animals infused with Emetine after "all cues" condition had selective reduction for the discrimination of the object associated with AC during the training session (Fig 2E). This indicates that the effect of Emetine on memory reconsolidation is specific for the object-context association because only the trace that was previously associated with the presented context was sensitive to Emetine infusion. Additionally, this suggests that all the contexts by themselves are capable of guiding reactivation of an object memory originally associated to that context, leaving the memories of objects linked to a different context unaffected by the action of protein synthesis inhibitors.

## CA3 place cells remapping correlates with memory recall

While several studies have shown that different experiences can be represented in the hippocampus [41,42], a direct link between the individual's internal context representation and memory retrieval is still missing. Taking advantage of our new behavioral paradigm, which allows us to measure how well animals remember a given context, we aimed to address an important yet unresolved question: How is place cell coding (in particular remapping) related to spatial memory? To assess the relationship between contextual representation and object in context memory, we recorded CA3 activity as animals performed the 1-day version of the task (4 rats, 24 sessions, $n = 332$ place cells). We compared place cell firing properties between training and test phases using parameters that describe similarities and differences between their place fields in both phases of the task, like the spatial correlation, and firing rate change (Fig 3, see also Methods section). We found that a place cell can maintain the position where they fire (similar location of its place field, high spatial correlation) and its firing rate (low firing rate change) or change one or both variables at the same time (Figs 3A and S5) without changing its spatial information coding, the size of their place fields or the proportion of place cells coding the environment (S6A–S6C Fig). When we grouped place cells according to the test condition, we found that the spatial correlation differed significantly between the NC and the other 2 conditions (Fig 3C, $n = 54–141$, Kruskal–Wallis test NC-AC $p = 0.0048$, NC-PC $p = 0.0108$, and AC-PC $p = 0.8523$), but not the firing rate change (Fig 3D, $n = 54–141$, Kruskal–Wallis test NC-AC $p = 0.6012$, NC-PC $p = 0.1467$, and AC-PC $p = 0.4754$). Interestingly,

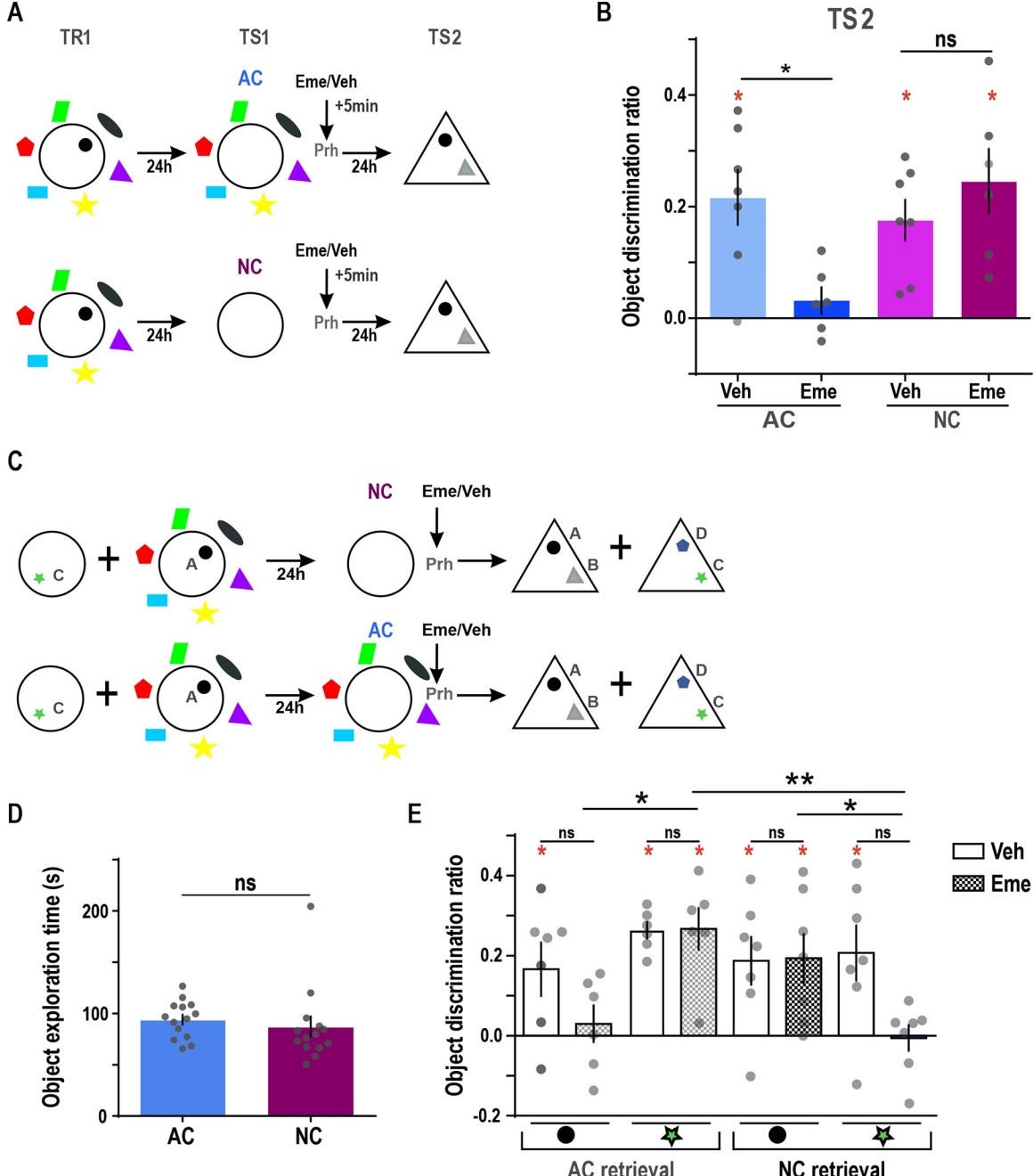

**Fig 2. Contextual cues are used to guide retrieval of the associated object memory trace in the Prh. (A)** Procedure and time points of infusions. Animals were trained in the task and 24 h after were exposed to an empty AC or NC context (TS1) and immediately after they received an Emetine (Eme) or Vehicle (Veh) infusion in the Prh. A test session was given to evaluate the original object against a novel one (TS2). **(B)** Discrimination ratio for the TS2 test session, 24 h after exposure to an empty context with AC or NC followed by an Emetine (dark) or Veh (light) infusion. RM two-way ANOVA: $F_{interaction} = 6.25$, $p = 0.029$, $F_{condition} = 2.90$, $p = 0.117$, $F_{drug} = 2.50$, $p = 0.142$, $n = 6-7$, Veh vs. Eme AC $p = 0.014$. NC $p = 0.479$. One sample $t$ test against 0: NC-Veh t = 4.77, $p = 0.003$; NC-Eme t = 4.21, $p = 0.008$; AC-Veh t = 4.38, $p = 0.005$; AC-Eme t = 1.30, $p = 0.252$. ACVeh: 0.22 ± 0.05, ACEme: 0.03 ± 0.02, NCVeh: 0.18 ± 0.04, NCEme: 0.24 ± 0.06. **(C)** Experimental protocol and time points of the Emetine or Vehicle infusion. Animals were trained to an object A (circle) in the NC context and an object C (star) in the NC context. The next day, they received an exposure session to an empty AC or NC context (no object) and immediately afterward they were infused with either Emetine (dark) or Vehicle (light) in the Prh. Lastly, object memories were tested against novel objects in a different context 24 h after (TS2). **(D)** Object exploration time during training in the AC or NC context. Mann–Whitney test U = 60 $p = 0.089$. AC: 94.08 ± 4.88, NC: 87.00 ± 10.11. **(E)** Discrimination ratio for the A or C objects against a novel object during TS2, 24 h after exposure to an empty AC or NC context followed by Emetine (gridded) or Vehicle (smooth) in the Prh. Two-way ANOVA: $F_{interaction} = 2.08$, $p = 0.132$, $F_{condition} = 4.05$, $p = 0.020$, $F_{drug} = 4.77$, $p = 0.039$, $n = 6-7$. AC

retrieval: CircleVeh: 0.17 ± 0.07, CircleEme: 0.03 ± 0.05, StarVeh: 0.26 ± 0.02, StarEme: 0.27 ± 0.05. NC retrieval: CircleVeh: 0.19 ± 0.06, CircleEme: 0.19 ± 0.06, StarVeh: 0.20 ± 0.07, StarEme: −0.01 ± 0.03. Red * represents significance against 0, # $p < 0.1$, * $p < 0.05$, ** $p < 0.01$. Individual values used to calculate mean and SEM are presented as dots. The data that support these findings is available in OSF at https://osf.io/7pw23/. AC, all cue; NC, no cue; RM, repeated measures.

even after removing half of the cues (PC), there were no significant differences in the spatial correlation or firing rate change of place cells when compared with the AC condition (Fig 3C and 3D). This phenomenon cannot be explained by differential context exploration, as the animals show no significant differences between conditions in the distance traveled, in the mean instantaneous velocity, or in the comparison of time spent in each spatial bin between phases (S5 Fig). These neuronal results are consistent with the behavioral output observed in the PC condition where behavior was not different between PC and AC (Fig 1G). Crucially, this suggests that retrieval of the original contextual memory in the PC context is directly associated with CA3 neuronal representation by place cells.

One advantage of the present task is its variability. We could find sessions in which, despite having reduced the number of cues (NC or PC condition), animals showed low percentage of exploration and sessions in which animals exhibited high percentage of exploration under a degraded context. This behavioral characteristic was useful to look for a relationship between place cell activity and the memory output (represented as the object exploration percentage) independently of the experimental condition. There was a significant inverse correlation between spatial correlation and percentage of object exploration and a significant positive correlation between firing rate change and percentage of object exploration. Thus, animals tend to explore the object more as firing rate change in CA3 increases and spatial correlation decreases (S6D and S6E Fig, spatial corr $p = 1.47\text{e-}10$ R = −0.34 and firing rate change $p = 0.00015$ R = 0.20).

According to dichotomic accounts of memory retrieval, animals' behavioral output should be represented by only 2 distributions, i.e., retrieval of the prior experience/high place cell activity correlation or no retrieval/low place cell activity correlation [43–45]. To understand if our behavioral data fitted this dichotomic account of memory, we performed a bootstrap likelihood ratio test (LRT) for assessing the number of mixture Gaussian components that could model our behavioral data. We found that the addition of a second distribution significantly increased the ability of the mixture Gaussian component model to explain our data (LRT, $p = 0.001$), but adding a third Gaussian component did not lead to a significant improvement (LRT, $p = 0.997$). This suggests that the behavioral output of our animals can be modeled by 2 independent distributions (i.e., memory retrieval/no memory retrieval, S7A Fig). We then classified the recording sessions in terms of the animal's dichotomic behavioral output (i.e., putative internal context representation) instead of the experimental cue setup (AC, PC, or NC). We separated sessions in which animals discriminated between contexts (high exploration, consistent with non-retrieval) from sessions in which they did not (low exploration, consistent with retrieval), using an estimated 2-SD threshold for context discrimination (see Methods, Fig 3E). With this new classification, we found significant differences in the spatial correlation (Fig 3F, $n = 65–267$, LMM $p = 1.822\text{e-}06$) 3G, $n = 65–267$, LMM $p = 0.0038$). Similar results were observed when comparing only neurons from the NC condition with different putative internal context representations (S7B and S7C Fig). Overall, place cell activity seems to better represent animal's internal representation than just the physical properties of a context. To rule out nonspecific effects, we repeated the same analysis but using the activity of neurons that were not classified as place cells and found no differences between groups (S7D Fig). In addition, we calculated the percentage of time the animals explore an area similar in

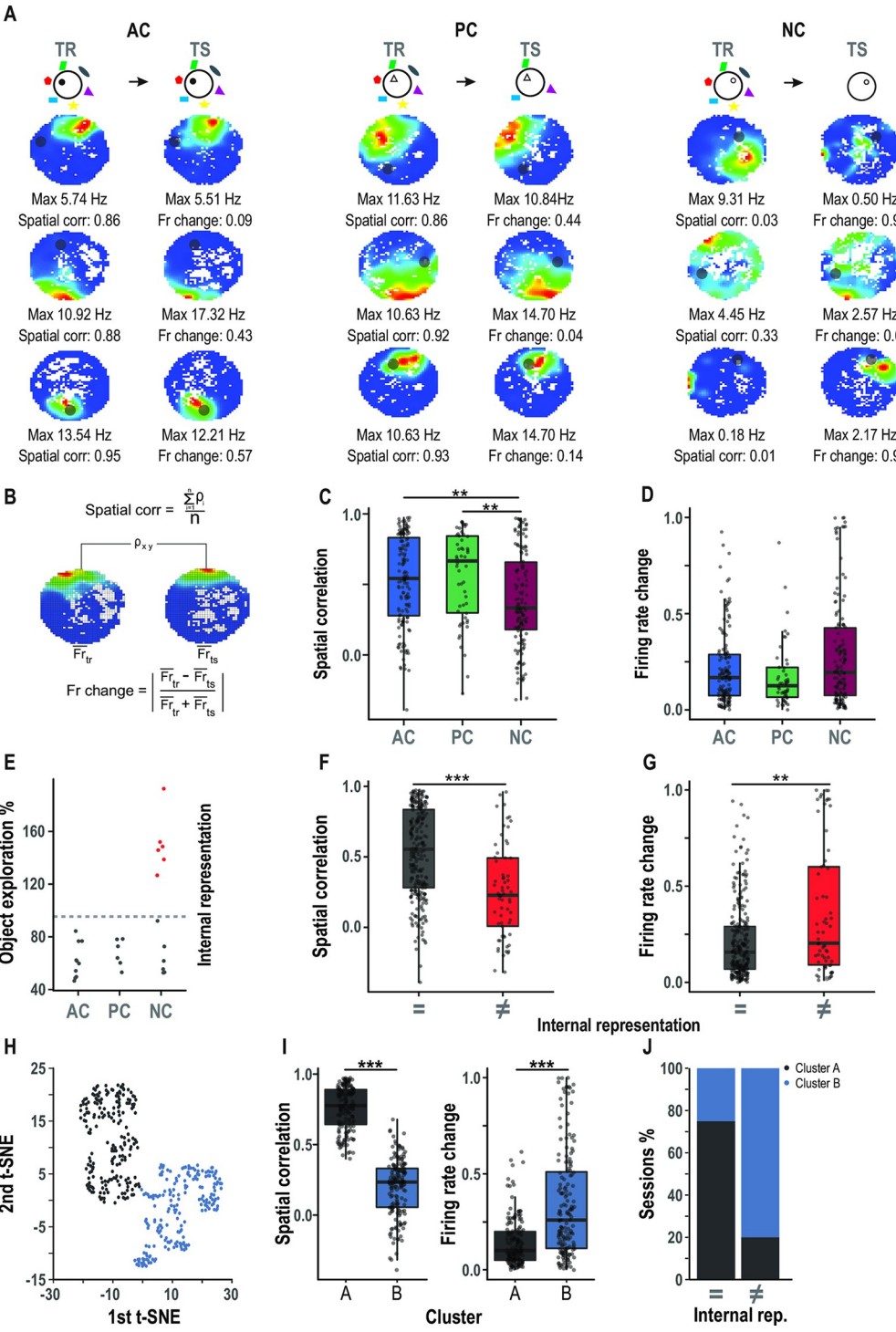

**Fig 3. CA3 place cells remapping correlates with memory recall.** **(A)** CA3 place cell responses. Place maps for training session (left) and test session (right). Color-coded firing maps normalized by the min and max firing rate of each neuron. Spatial correlation and firing rate change were calculated for each neuron. **(B)** Schematic of the calculation of spatial correlation and firing rate change. **(C)** CA3 spatial correlation sorted by condition. Only in NC sessions, place cells had a significantly lower spatial correlation. Moreover, there were no differences between the AC and PC conditions. $n = 54–141$, Kruskal–Wallis test NC-AC $p = 0.0027$, NC-PC $p = 0.0074$, and AC-PC $p = 0.8503$. AC: $0.53 ± 0.03$, PC: $0.56 ± 0.04$, NC: $0.39 ± 0.03$. **(D)** CA3 firing rate change sorted by condition. There are no differences between conditions. $n = 54–141$, Kruskal–Wallis test NC-AC $p = 0.3430$, NC-PC $p = 0.0720$, and AC-PC $p = 0.4788$. AC: $0.22 ± 0.02$, PC: $0.19 ± 0.03$, NC: $0.29 ± 0.02$. **(E)** Object exploration percentage for every condition and

for the internal representation: sessions were divided by a context discrimination threshold (mean (OE% AC OE% PC) + 2 STD)) red dots represent sessions where animals discriminated between contexts and gray dots, sessions where animals did not discriminate. **(F)** CA3 spatial correlation sorted by putative internal representation. $n$ = 65–267, LMM $p$ = 2.311e-06, =: 0.53 ± 0.02, ≠:0.25 ± 0.04. **(G)** CA3 firing rate change sorted by internal representation. $n$ = 65–267, LMM $p$ = 0.006, =: 0.21 ± 0.01, ≠:0.36 ± 0.04. **(H)** t-SNE over the spatial correlation and firing rate change and K-means clustering, gives 2 clear clusters evaluated by the Silhouette method. **(I)** Spatial correlation and firing rate change sorted by cluster. Cluster A has higher mean spatial correlation and low firing rate change than cluster B. $n$ = 165–167, LMM, p spatial corr. = 1.9186e-63 (A: 0.76 ± 0.01, B: 0.20 ± 0.02) and p firing rate change = 1.0141e-14 (A: 0.14 ± 0.01, B: 0.34 ± 0.02). **(J)** Percentage of sessions with majority of cluster A or B sorted by putative internal representation. X2 (1, $N$ = 21) = 4.887, $p$ = 0.0271. The data that support these findings is available in OSF at https://osf.io/7pw23/. AC, all cue; NC, no cue; PC, partial cue; t-SNE, t-distributed stochastic neighbor embedding.

size to the object and opposite to it, and sorted place cell's activity according to this new behavioral output. We found no differences between groups for the 2 studied variables (S7E Fig).

To better understand how the neuronal population might represent this dichotomic behavioral output, we examine whether our neuronal population can be classified according to its spatial correlation and firing rate change values. To do this, we computed t-SNE (t-distributed stochastic neighbor embedding) over the 2 variables and performed a k-mean clustering, finding 2 clear clusters (evaluated by the Silhouette method). Cluster A has a higher mean spatial correlation than Cluster B (Fig 3I, $n$ = 165–167, LMM, $p$ = 1.9186e-63) and a lower firing rate change (Fig 3I, $n$ = 165–167, LMM, $p$ = 1.0141e-14). Looking at the proportion of each cluster according to the internal representation, we found that Cluster B is more represented in the non-retrieval session (Fig 3J, 80%, 4 out of 5) and Cluster A is more represented in the retrieval sessions (Fig 3J, 75%, 12 out of 16, X2 (1, $N$ = 21) = 4.887, $p$ = 0.0271). In addition, most cells from the session where animals had a different internal representation (79%) are in Cluster B. This suggests that Cluster B grouped neurons represent a change in the animal's internal representation. In summary, when animals are unable to recall the initial memory, there is a greater representation of cells from Cluster B. These cells exhibit a low spatial correlation and a high firing rate change.

The differences in the spatial coding between phases of the task were also analyzed using a population vector analysis (S7C Fig). This analysis showed the same tendency as the one observed at the level of single units, where the spatial coding is more stable in non-retrieval sessions. We also fitted our neuronal data with a general linear model (GLM) with a binomial distribution, using the putative internal context representation as the response variable. Models including only one of the variables (spatial correlation or firing rate change) or both were significant (internal rep ~ spatial corr, $p$ = 1.27e-08; internal rep ~ firing rate change, $p$ = 1.63e-05 and internal rep ~ spatial Corr + firing rate change, $p$ spatial corr = 1.97e-06, $p$ firing rate change = 0.02). The best of the 3 models is the one that includes both variables (AIC 1-variable model = 294.30 spatial corr, 314.01 firing rate change, AIC 2-variables model = 291.16, and ANOVA comparing models with increasing variables, $p$ 1-variable model versus $p$ 2-variables model = 0.009). Although the firing rate change appears to be a less reliable estimator of the animal's internal representation compared to spatial information, it still provides significant information, as demonstrated by the t-SNE analysis and the GLM model.

From these results, we can conclude that place cell population encode more than the physical space; instead, their activity is related to the memory that animals express in each context.

## Role of NMDAR during associative retrieval of object in context memories

We have demonstrated a correlation between CA3 place cell remapping and retrieval memory in our behavioral paradigm. It is well-established that the generation and modification of place fields are related to NMDAR-activity [46–51]. Furthermore, previous studies have as well

shown that the CA3 region [24,52], and NMDARs in particular, are required when retrieval occurs in the absence of some of the original contextual cues [28]. To assess the requirement of CA3-NMDARs for retrieval under cue-degraded conditions, we used a pharmacological approach. Animals trained in the 2-day version of the task received an injection of the NMDAR antagonist AP5 or Vehicle in the CA3 region 15 min before the test session in the AC or PC condition (Fig 4A). Vehicle-infused animals spent significantly less time exploring the original object, independently of the condition. However, animals that received an AP5 infusion prior to the test session in the PC condition showed significantly higher exploration percentages than Vehicle infused animals. In contrast, no effect of AP5 was observed in the AC condition. In accordance with this, the proportion of animals that show high exploration (Fig 4B, consistent with non-retrieval in the test, in red) varies from 18% in "all cues" and 0% "partial cues" in vehicle injected sessions to 50% in the "partial cues" AP5 injected test sessions (Fig 4C). Our results suggest that the NMDAR activity in the CA3 region is involved in recovering memories in the presence of partial contextual information but not with all the original cues.

Then, we asked whether NMDARs were required for reactivation of the object-context memory under degraded cues. For this, after training on a context-object association as in previous experiments (day 1), animals infused with either AP5 or Veh in CA3 were subjected to an "all cues" contextual re-exposure in the absence of any object, and Emetine or Vehicle was infused in the Prh immediately after (TS1). On day 3, the original object memory of day 1 was tested on the triangular context against a novel object (TS2). Consistent with our previous result (Fig 2), Emetine infusion in the Prh immediately after context exposure (TS1) significantly reduced the discrimination ratio of the previously presented object even under a cue-degraded context (Fig 4D). But when we combined pre-TS1 infusion of AP5 into the CA3 with post-TS1 infusion of Emetine into the Prh, Emetine only reduced the discrimination ratio of the object memory under AC but not under PC context exposure, suggesting that inhibiting NMDAR activity during PC exposure can interfere with the ability to activate the original memory representation. The PC specificity of AP5-effect during the exposure session rules out an effect due to changes in memory labilization induced by AP5. Furthermore, the absence of any significant difference in the number of rearings exclude possible AP5-induced changes in exploration levels during the exposure session (paired $t$ test: AC Mean ± SEM Veh = 36.65 ± 5.84 AP5 = 42.53 ± 32.77, $t$ = 0.78, $p$ = 0.444, PC Mean ± SEM Veh = 56.9 ± 4.37 AP5 = 53.05 ± 4.89, $t$ = 0.84, $p$ = 0.409). In sum, these results showed that NMDARs in the CA3 region are not required for memory reactivation when guided by the original contextual information, but are crucial for the reactivation of object-in-context memories under partial contextual information when a memory completion process is required.

## Bidirectional modulation of NMDAR activity in the DG-CA3 circuit affects the balance between memory retrieval and acquisition under cue-degraded conditions

Many findings indicate the importance of NMDAR in the HP (especially the DG) for correct pattern separation function, both at the behavioral [25] and electrophysiological level [26,49–51]. Interfering with NMDARs could hinder memory differentiation thereby increasing memory generalization. To address this, we tested the role of NMDAR activity in the DG for retrieval under the AC, PC conditions, and an additional PC2 condition in which only 2 cues were left (Fig 5A). We found no effect of AP5 infusions in the DG 15 min previous to the test session in the AC or PC condition (Fig 5B). However, using a two-cue context (PC2) during test session, we observed that DG-Vehicle-infused animals showed percentages of exploration that did not differ significantly from training, while AP5 infused animals had a percentage of

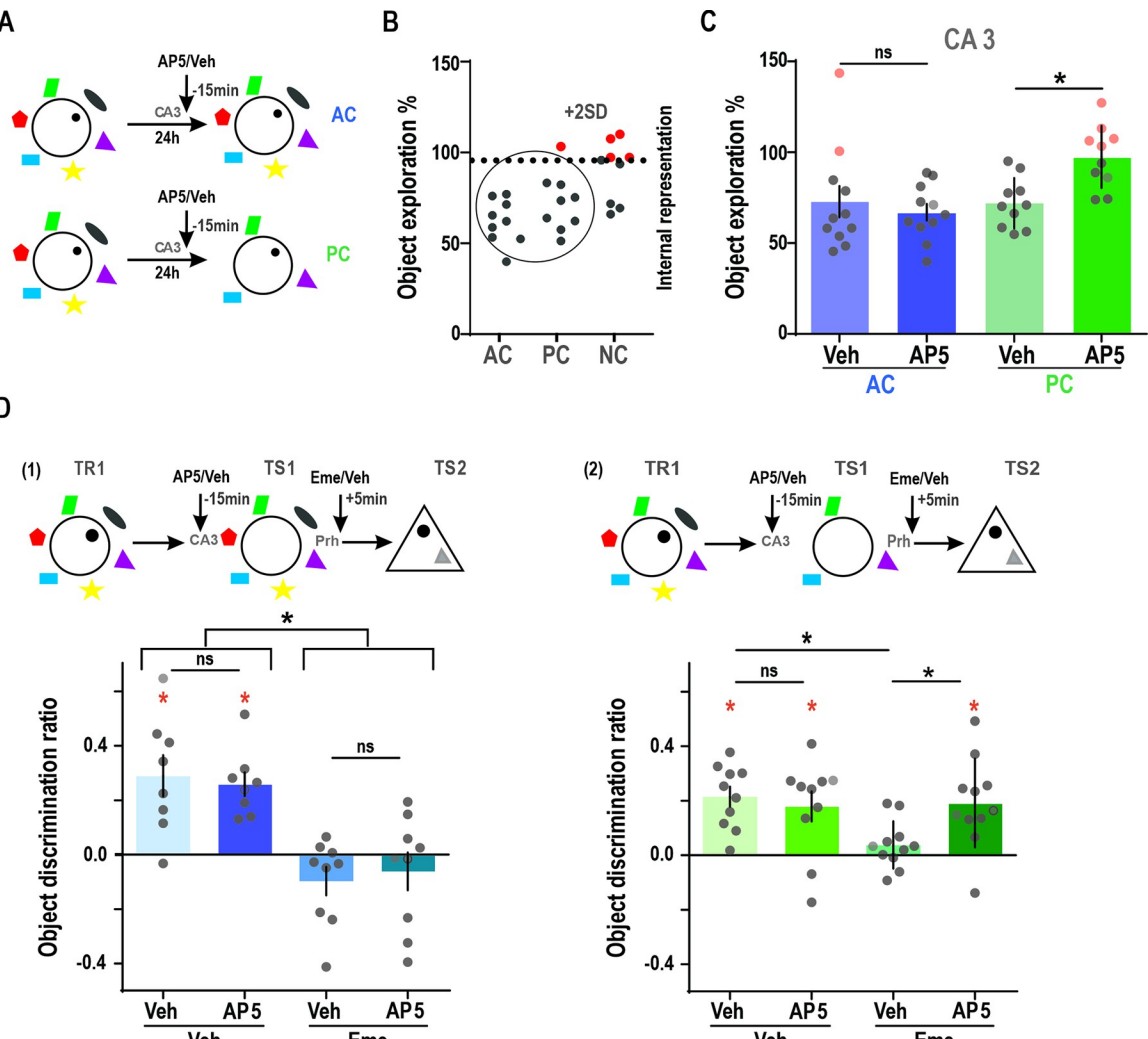

**Fig 4. NMDAR requirement for associative retrieval of object in context memory in a cue-degraded context. (A)** Time of infusion of AP5 or Vehicle (Veh) in CA3 15 min before test session in the AC and PC conditions. **(B)** Data from PC and TC groups from Fig 1D was used to obtain the 2SD threshold criteria for discriminating sessions where putative internal context representations would be consistent with non-retrieval, as done in Fig 3. **(C)** Percentage of exploration for the object presented in the presence of the AC or PC conditions during test session, in animals infused with Vehicle (light) or AP5 (dark) in the CA3 region of the HP. Two-way ANOVA: $n = 10–11$, $F_{interaction} = 4.50$, $p = 0.047$, $PC_{Veh-AP5}$ $p = 0.042$, $AC_{Veh-AP5}$ $p = 0.887$. Wilcoxon test against 100%, AC-Veh W = −48, $p = 0.032$; PC-Veh W = −66, $p = 0.001$; AC-AP5 W = −55, $p = 0.002$; PC-AP5 W = −7, $p = 0.769$. $AC_{Veh}$: 72.85 ± 8.64, $AC_{AP5}$: 67.02 ± 4.59, $PC_{Veh}$: 71.98 ± 4.39, $PC_{AP5}$: 97.42 ± 5.42. **(D) (Up)** Trained animals were infused with AP5 or Veh in CA3 before test session in an AC (1) or PC (2) conditions and infused with Veh or Emetine (Eme) in the Prh immediately after test session (TS1); 24 h after, memory for the original object was tested in another familiar context against a novel object (TS2). **(Down)** (1) Discrimination ratio for the TS2 session of the task, 24 h after exposure to an empty original context followed by an Emetine or Vehicle infusion in animals that had previously received AP5 or Vehicle infusions. Two-way ANOVA: $F_{interaction} = 0.26$ $p = 0.618$, $F_{Veh-Eme} = 35.93$ $p < 0.0001$, $F_{Veh-AP5} = 0.001$ $p = 0.970$, $n = 8–9$. One sample $t$ test against 0, Veh-Veh t = 3.81, $p = 0.007$; AP5-Veh t = 5.98, $p = 0.0006$; Eme-Veh t = 1.85, $p = 0.101$; Eme-AP5 t = 0.89, $p = 0.399$. $AC_{Veh/Veh}$: 0.29 ± 0.08, $AC_{AP5/Veh}$: 0.26 ± 0.04, $AC_{Veh/Eme}$: −0.10 ± 0.05, $AC_{AP5/Eme}$: −0.06 ± 0.07. (2) Discrimination ratio for TS2 session of the task, 24 h after exposure to an empty partial cue context followed by an Emetine or Vehicle infusion in animals that had previously received AP5 or Vehicle infusions. Two-way ANOVA: $F_{interaction} = 4.42$ $p = 0.049$, $n = 10–11$; Veh/Veh vs. Veh/AP5 $p > 0.99$, Veh/Eme vs. Eme/Veh $p = 0.046$. One sample $t$ test against 0: Veh-Veh t = 5.83, $p = 0.0003$; AP5-Veh t = 3.23, $p = 0.010$; Eme-Veh t = 1.44, $p = 0.181$; AP5-Eme t = 3.90, $p = 0.003$. $PC_{Veh/Veh}$: 0.21 ± 0.04, $PC_{AP5/Veh}$: 0.18 ± 0.06, $PC_{Veh/Eme}$: 0.04 ± 0.03, $PC_{AP5/Eme}$: 0.19 ± 0.05. Red * represents significance against 0, * $p < 0.05$, *** $p < 0.001$. Individual values used to calculate mean and SEM are presented as dots. Red dots represent animals whose object exploration was above the memory retrieval threshold (i.e., putative internal context representation). The data that support these findings is available in OSF at https://osf.io/7pw23/. AC, all cue; PC, partial cue.

exploration significantly lower than 100% and significantly different from the Vehicle-infused group. Consistent with this, the proportion of animals showing non-retrieval behavior in test sessions in the PC2 (in red) goes from 60% in Veh injected sessions to 20% in the AP5 injected sessions (Fig 5C). The lack of effects in the less degraded AC and PC conditions rules out any nonspecific influence of AP5 on exploration levels. Our results suggest that NMDARs in the DG are necessary for the orthogonalization that allows a distinction to be formed between a cue-degraded context and the original full context, and that the absence of this distinction could lead to memory retrieval even in highly degraded contexts.

However, we predicted that we could favor memory discrimination by activating NMDAR in the DG. We tested this prediction in animals trained in the 2-day version of the task that received an infusion of the NMDAR partial agonist D-Cycloserine (D-Cyc) or Vehicle in the DG 15 min before exposure to a PC context (Fig 5D). Animals infused with D-Cyc in the dentate gyrus before the test session showed significantly higher exploration percentages than the vehicle group, with object exploration times that did not differ from the training session. In line with this, the proportion of animals that show high exploration consistent with non-retrieval in test sessions (in red) goes from 12% in Veh injected sessions to 37% in the AP5 injected sessions (Fig 5E). It is important to note that animals infused with D-Cyc showed no increase in the distance traveled or linear velocity during the test session compared with Veh injected animals. This outcome rules out an effect of D-Cyc over general mobility levels (S3 Fig). All in all, these results indicate that NMDAR activity in the DG can decrease the ability to generalize from a degraded context. In contrast, we have previously shown that NMDARs in the CA3 region are important for contextual generalization. We reasoned that by increasing NMDAR activity in CA3, we could increase retrieval guided by incomplete cue stimuli. We infused D-Cyc or Vehicle in CA3 before the test phase under the PC2 context and found that Veh-infused animals showed exploration levels consistent with a lack of object in context memory retrieval (Fig 5D and 5F). However, the infusion of D-Cyc in the CA3 previous to this test phase significantly decreased the percentage of object exploration. Consistently, the percentage of animals that show high exploration (i.e., non-retrieval) goes from 56% to 22% (Fig 5F). This indicates that increasing NMDAR activity in CA3 can enhance memory retrieval under cue-degraded conditions, potentially through tipping the balance towards contextual generalization.

Changes in engram cell excitability [53] or plasticity [54–56] could be controlling the efficacy of memory retrieval under contextually degraded conditions in our task. In this regard, NMDA receptor activation can trigger AMPA-type trafficking to the dendritic surface in an L-type voltage gated calcium channel (L-VGCC)-dependent manner [57] and L-VGCC can also regulate neuronal excitability [58,59]. Therefore, we evaluated their role as potential effectors of NMDARs. Animals were infused in CA3 with the L-type calcium channel inhibitor Nimodipine (Nim) prior to the D-Cyc infusion and then tested in the PC2 context. While D-Cyc infusion decreased the percentage of object exploration, co-infusion of D-Cyc with Nimodipine prevented this effect (Fig 5G). This result suggests that NMDARs interact with L-VGCC in the CA3 region to favor memory generalization. Histological analysis did not reveal any lesion in the infusion sites (CA3 or DG) (S2 Fig). Results cannot be explained by drug-related changes in motivation to explore the object during the test phase, since infusion of D-Cyc in the DG or CA3 region 15 min before exposure to an object in the AC context did not alter exploration percentages compared with Vehicle-infused groups (paired $t$ test: $t_{DG} = 0.67$, $p = 0.528$, $n = 7$; $W_{CA3} = -8$, $p = 0.578$, $n = 7$).

In summary, our results suggest that bidirectional manipulation of NMDARs in CA3/DG affects the hippocampal processing balance between a retrieval mode (favored by generalization-like processing) and an acquisition mode (favored by a differentiation-like processing).

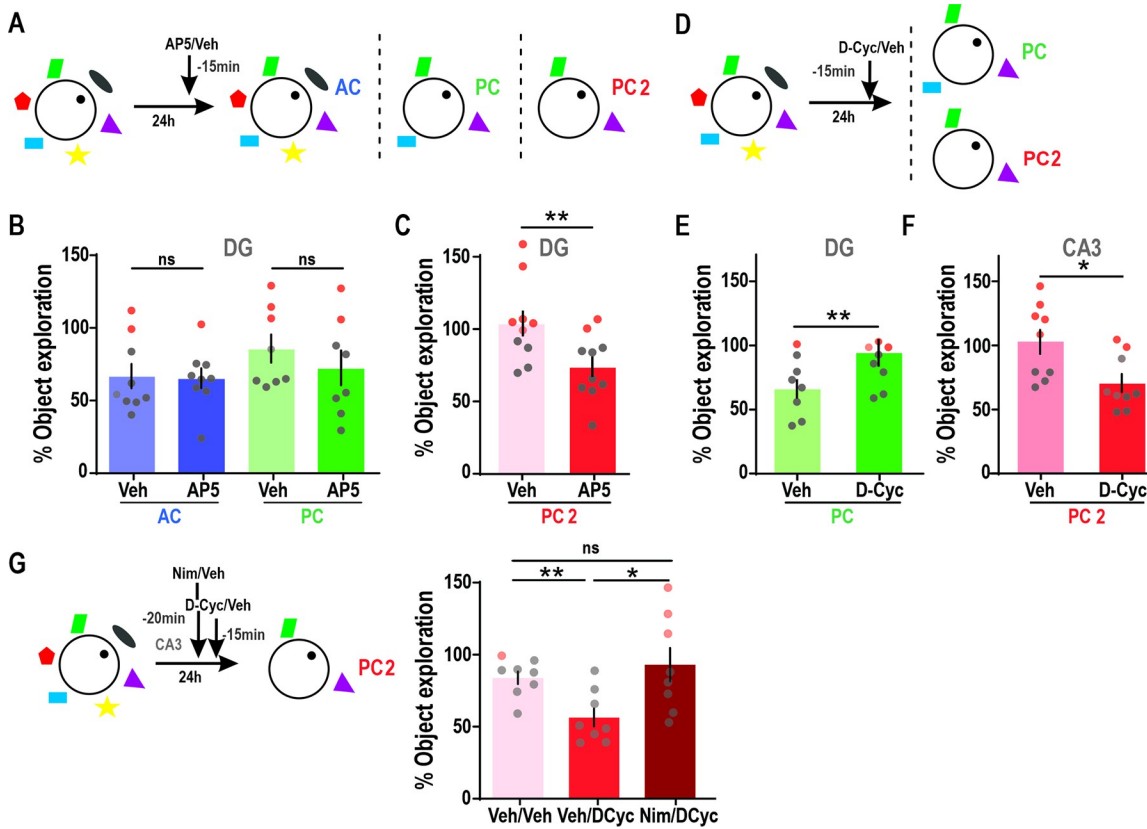

**Fig 5. NMDAR agonist in CA3 increases retrieval capacity in the presence of partial cues, in an L-type voltage-gated calcium channel-dependent manner, while in DG has an opposite effect. (A)** Time of infusion of NMDAR agonist D-Cyc or Vehicle (Veh) in the DG 15 min before the evaluation session in the AC, PC, or PC2 condition of the task. **(B)** Percentage of exploration for objects presented during the test session under the AC or PC conditions of the task in animals infused with either Vehicle (light) or AP5 (dark) in the DG. Two-way ANOVA GD: $n = 8$–9, Finteraction = 0.42, $p = 0.527$, Fcondition = 1.93, $p = 0.185$, Fdrug = 0.65, $p = 0.432$. One sample $t$ test against 100%, Veh AC t = 3.94, $p = 0.043$; AP5 AC t = 5.05, $p = 0.001$; Veh PC t = 1.467, $p = 0.186$; AP5 PC t = 2.30, $p = 0.055$. ACVeh: 66.83 ± 8.42, ACAP5: 65.37 ± 6.86, PCVeh: 85.76 ± 9.71, PCAP5: 72.51 ± 11.95. **(C)** Percentage of exploration in the PC2 context during test session in Vehicle infused (light) or AP5 infused animals (dark). Paired $t$ test t = 3.26, $p = 0.010$, $n = 10$. One sample $t$ test against 100%, Veh $p = 0.669$, t = 0.44; AP5 $p = 0.005$, t = 3.65. PC2Veh: 103.9 ± 8.89, PC2AP5: 74.09 ± 7.09. **(D)** Time of infusion of D-Cyc or Vehicle in the DG or CA3 15 min before test session in the PC or PC2 condition. **(E)** Percentage of object exploration during test session in PC condition of the task in animals infused with Veh (light) or D-Cyc (dark) in DG prior to the session. Paired $t$ test: t = 4.198, $p = 0.004$, $n = 8$. One sample $t$ test against 100%: Veh t = 3.90, $p = 0.006$; D-Cyc t = 2.54, $p = 0.04$. PC$_{Veh}$: 68.36 ± 8.10, PC$_{DCyc}$: 84.80 ± 5.98. (F) Percentage of exploration during test session in PC2 condition of the task after animals were infused with Veh (light) or D-Cyc (dark) in CA3. Paired $t$ test: t = 3.03, $p = 0.016$, $n = 9$. One sample $t$ test against 100%, Veh t = 0.31, $p = 0.767$; D-Cyc t = 4.15, $p = 0.003$. PC$_{Veh}$: 103.00 ± 9.69, PC$_{DCyc}$: 70.65 ± 7.08. **(G)** (Left) Time of Nimodipine (Nimo)/Veh infusion and D-Cyc/Vehicle in CA3 20 and 15 min, respectively, before test session. (Right) Percentage of exploration during test session in PC2 condition of the task after animals were infused with Veh/Veh (light) or Veh/D-Cyc (red) or Nim/D-Cyc (dark) in the CA3 region. RM one-way ANOVA: F = 5.44, $p = 0.037$, $n = 8$. * $p < 0.05$, ** $p < 0.01$. Veh/Veh t = 3.62, $p = 0.0085$, Veh/D-Cyc t = 6.717, $p = 0.0003$; Nimo/D-Cyc t = 0.61, $p = 0.5668$. PC2$_{Veh/Veh}$: 84.30 ± 4.34, PC2$_{Veh/DCyc}$: 56.64 ± 6.45, PC$_{Nim/DCyc}$: 93.01 ± 11.63. Individual values used to calculate the mean and SEM are presented as dots. Red dots represent animals whose object exploration was above the memory retrieval threshold (i.e., internal context representation). The data that support these findings is available in OSF at https://osf.io/7pw23/. AC, all cue; PC, partial cue; RM, repeated measures.

Finally, we performed a dichotomic analysis to account for the putative effect of these drugs on memory retrieval, similar to that used in the electrophysiological experiments (i.e., putative internal context representation). We divided the data into 2 groups by pooling together all "memory boosting" (left) or "memory impairing" (right) drug infusion experiments (see Methods, Fig 6A). We found that in the "memory boosting" group, in a two-cue context (where most animals show a behavior inconsistent with retrieval), the treatment increased the

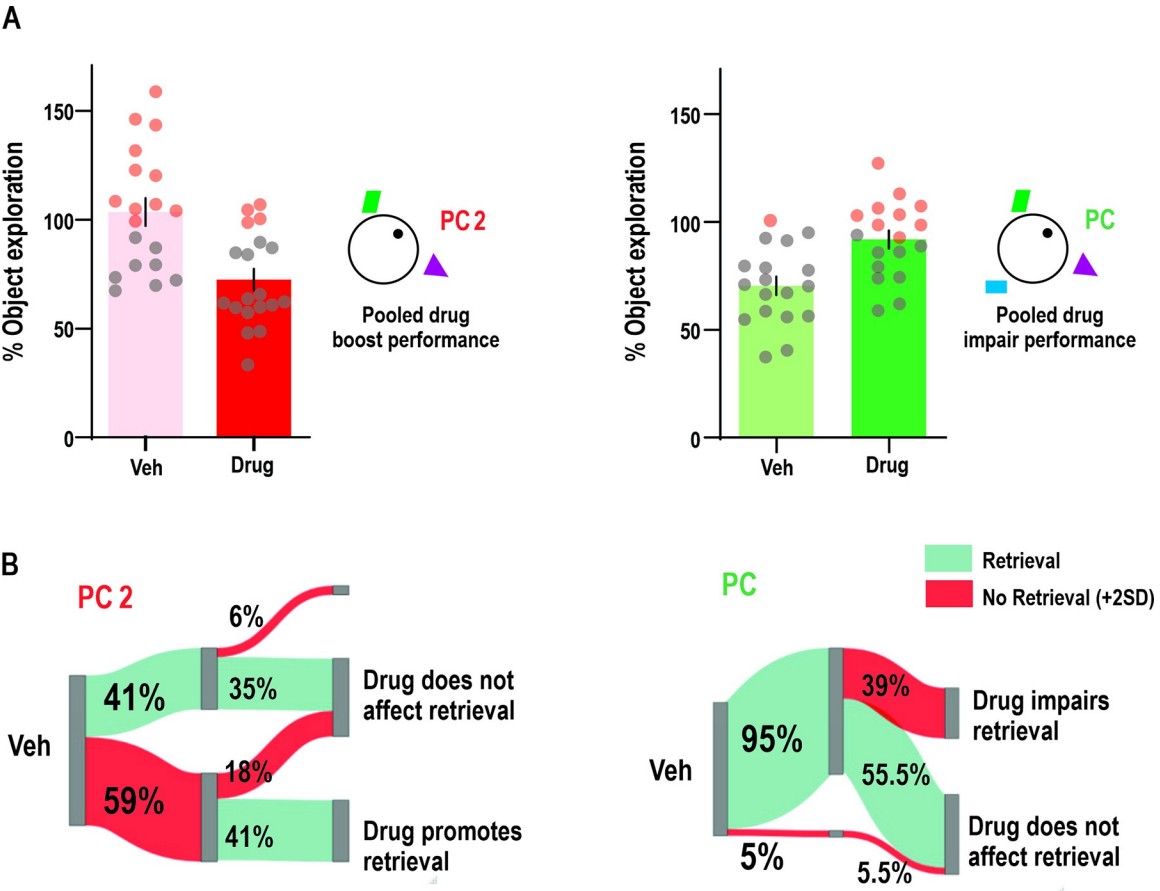

**Fig 6. Drugs affect animals' context representation in a cue-degraded environment. (A)** Pooled distribution of all "memory boosting" (left) or "memory impairing" (right) drug infusion experiments, depicting as red dots the animals above the memory retrieval threshold (i.e., putative internal context representation). **(B)** Percentage of animals that were above 2SD retrieval threshold (red) during Veh session or below retrieval threshold (green), showing how much of each percentage is affected by the drug or remains unchanged. (X2 = 3.125, $p$ = 0.03855, pool drug impair performance, X2 = 5.14286, $p$ = 0.0167, one tail McNemar chi-square test). Pooled Boost, Veh: 103.5 ± 6.37, Drug: 72.46 ± 4.89, Pooled impair, Veh: 70.37 ± 4.23, Drug: 91.81 ± 4.18. The data that support these findings is available in OSF at https://osf.io/7pw23/. PC, partial cue.

proportion of rats that behaved in a retrieval-like manner (similar internal context representation between test and training, Fig 6B, left). Conversely, in the "memory impairing" group, in a three-cue context where most animals behave in a retrieval-like fashion, the treatment decreased the number of rats behaving in a retrieval-like manner (Fig 6B, right). These results show that, although there is a great variability in animal behavior, NMDAR activity in the hippocampal circuit can bias the internal spatial representation towards memory retrieval or away from it.

## Discussion

In this study, we investigated the molecular and neural mechanisms involved in the recovery of spatial representations stored as incidental memories of an object experienced in a particular context. Our findings indicate that memory differentiation and memory generalization functions compete for behavioral control. We showed that the amount of CA3 remapping, signaling a separation process, is related to the recall of object-in-context memory. When animals recognized an object in context as familiar, regardless of the available contextual cues, there

was less remapping in CA3 than when animals recognized the relationship as novel. This finding is crucial for understanding how contextual representations can influence episodic memory recall and behavior. In the same line, we showed that NMDAR activity in the DG-CA3 circuit can influence this discrimination/generalization balance. We found that pharmacological treatments that favor pattern separation (CA3 NMDAR antagonist and DG NMDAR agonist) lead animals to behave as if the relationship between object and context is novel while treatments that promote pattern completion (CA3 NMDAR agonist and DG NMDAR antagonist) bias animals to retrieve a prior experience.

Holistic memory retrieval is considered a key element of episodic memory allowing all aspects of an event to be recovered in an integrated manner. Our results indicate that animals can retrieve a memory of an object in a particular context, guided by a limited amount of contextual information, aligned with prior studies [24,28]. Furthermore, the similar patterns of object exploration and rearings in the presence of a variable number of retrieval cues indicate that exploration levels of familiar objects in the present task can be directly related to contextual novelty.

Place cells in the hippocampus can remap in response to contextual changes [60], enabling different activity patterns to represent different environments [61,62]. However, none of the previous studies have shown whether remapping has any behavioral significance (i.e., is there any link between remapping and memory?). There are interesting studies showing that artificial activation of a selected group of hippocampal neurons can modify mice behavior in a contextual fear memory [63–66] or a reward-oriented paradigm [67]. Here, we go a step further and show that the degree of spontaneous incidental memory, not involving reward or punishment, correlates with the amount of remapping of CA3 place cells. This result is fundamental for a better understanding of contextual representations and their role in memory, without the influence of explicit motivational factors.

Our results contain a large variability between (and within) animals in terms of how these contextual representations, coded by place cell activity, change with contextual variation. This has been repeatedly shown in the literature [19,68,69]. Place cells in CA1 can remap within the same session even though there are no changes in the context, possibly due to modifications in the internal state of the animal [19]. This variability could be explained if remapping does not solely reflect the observable and controlled properties of the environment, but rather subjective perceptions and predictions the animal is making about the environment during retrieval [70]. In fact, inter- and intra-individual variability in behavior is the rule rather than the exception [71–73]. Interestingly, we showed that the amount of memory retrieved was not only reliant on the test condition itself (number of cues) but was specifically related to the animal's internal representation of the context. Thus, sorting cell activity by animal's internal representation more clearly reflects the different encoding of distinct contexts for the animal. This contrast between the effect of the contextual manipulation (AC, PC, NC) and the actual change in contextual representations reveals that controlled experimental contextual manipulations only reflect a subset of what the animal perceives as context. Accordingly, individual representation of the same space is variable, but behaves in a dichotomous manner as expected according to the dual-process model of memory [43–45]. This representation can be more or less subject to discrimination or generalization depending on uncontrolled experimental variables that increase in importance as the context becomes more degraded. Our results contrast with those of Kentros and colleagues [20], who described an increase in the number and tuning of place cells when the number of environmental cues is higher. Importantly, that particular study recorded place cells from CA1, not CA3 like in the present manuscript, so those results are not easily comparable with ours. Having said this, in our task the 6 distal visual cues are likely not the only cues animals use to orient themselves in the arena, this could make the AC and NC

conditions more similar in terms of their contextual representation. This might be the reason why we were able to disentangle the internal representation from the external condition in the electrophysiology experiments. Despite the difference in the distal removable visual cues, there is a conserved ambiguity between the contexts that is reflected in place cell activity and also in behavior. In addition, the effect of the pharmacological manipulations over the generalization/ discrimination was also variable and likely dependent on the retrieval state of the animal (Fig 6B). We propose that the pharmacological treatments biased individual pattern completion/ pattern separation balance, leading to changes in the activation of the internal contextual representation during retrieval, and therefore modifying the distribution of exploration times towards discrimination or generalization. In the labilization/reconsolidation experiments, both contextual and object information are degraded. Thus, object memory retrieval demands a reconstruction of the original experience from degraded information to guide memory. Combined, our results show that incidental reactivation occurs for all the elements of the original event even in contextually degraded conditions, and that retrieval is related to a low degree of remapping in CA3 place cells. In addition, these results reveal a contextual specificity of the reactivation and subsequent labilization of object memory traces, since only the corresponding contextually associated object memory trace was reactivated under contextual exposure in the NC and AC retrieval sessions (Fig 2E). Consistent with this idea, human studies found that activity in the Prh correlates with recollection in response to partial spatial cues [74,75] and associative retrieval led to neocortical activity corresponding to incidental reactivation of all the items of a particular event [76].

Our results are consistent with other studies suggesting a requirement of NMDAR activity in the CA3 region for long-term spatial memory reactivation, but only when the amount of contextual information available is limited [28,77]. Since spatial place fields are generated and modified in an NMDAR-dependent manner [46–51], it is possible that memory reactivation under cue-degraded conditions (where presumably information input via the perforant pathway is reduced) relies more heavily on the activity of attractor states and the strength of recurrent collateral connections in CA3. Changes in NMDAR activity could affect the attractor circuit activation threshold since enhancement of CA3 recurrent collateral connections is NMDAR-dependent [55,56]. Therefore, NMDAR antagonists could be preventing reverberant activity in the CA3 region, leading to a retrieval deficit, and NMDAR agonists could favor attractor circuit stability for the corresponding memory trace, thereby promoting retrieval. Could memory retrieval under contextually degraded cues rely on LTP expression mechanisms that involve an NMDAR-dependent component in CA3? Although the role of NMDAR for LTP was mainly associated with induction [78,79], a recent publication showed that recall cues can drive transient increases in excitability of engram cells and this, in turn, can influence the efficacy of memory retrieval [53], pointing out a role of NMDAR in the expression of LTP. Interestingly, synaptic activation of NMDARs triggers internalization of Kir2.1 channels leading to increases in engram cell excitability. In our task, NMDAR-dependent increases in cell excitability in CA3, driven by the degraded context, could be controlling the efficacy of memory retrieval. Moreover, there is evidence that NMDARs could participate in both spatial memory and place field activity [28,50,80,81]. In addition, the fact that a voltage-gated calcium channel blocker (Nimodipine) prevented the increase in memory generalization due to CA3 D-Cyc infusion points toward a role of calcium channels and possible excitability changes regulating the strength of cue-driven retrieval under partial cue conditions.

Computational models suggest that the attractor circuit in CA3 could lead to pattern separation or pattern completion, as a function of the relative strength of the attractor circuit and the nature of the external inputs from the EC and DG [3,82,83]. Sensory information entering

through the perforant path could be used to either recover the original engram in CA3 and cause retrieval of the entire memory, or to form a new engram guided by the new entrance of information. Thereby, these theories imply that recognition of an experience will ultimately depend on whether pattern completion-like or pattern separation-like processes rule over CA3 activation patterns. This is exactly what we have seen in the electrophysiological data, in which the degree of remapping balances between retrieval of a familiar memory with low remapping or recognizing a different situation with no retrieval and higher remapping. In this context, we speculate that, under AC and PC conditions, input from the EC through the perforant path could lead to attractor activity in CA3 corresponding to an object-in-context representation and, by the time DG inputs arrive, the strong attractor dynamics of the circuit dominate and prevent remapping favoring a pattern completion process. But if the sensory inputs induced by retrieval cues are weaker, like in the PC2 condition, CA3 attractor circuit activity will also be weaker, offering the opportunity for these attractor states to follow DG input and promote environmentally induced remapping.

In this scenario, we have shown that interfering with plasticity-related mechanisms in the DG can actually favor memory retrieval in conditions where spatial cues would normally be insufficient to guide retrieval. In this sense, NMDAR inactivation could affect the ability of the DG to remap in response to novel stimuli [25], leading to stable memory representation in CA3 and memory retrieval under incomplete contextual information. Electrophysiological studies that support this line of thought have shown that LTP decay is an active process that requires NMDAR activity [84]. Consistent with the putative DG role in pattern separation, NMDAR partial agonist D-Cyc prevented memory retrieval in the PC condition, suggesting a shift in the hippocampal balance towards an acquisition mode instead of a retrieval mode. This is in line with the fact that DG granular cells are depolarized after exposure to a novel environment, indicating that these cells could be involved in setting the hippocampal circuit into an acquisition mode [85]. Moreover, although most studies focused on the role of NMDARs on spatial memory acquisition [86–89], a few studies already showed a post-acquisition facilitation by administration of NMDAR antagonists [84,90]. Although inactivation of NMDARs in the DG can modulate memory retrieval, it is likely that inactivation of NMDARs in the DG is not sufficient to support memory retrieval by itself. Rather, a requirement for external contextual inputs with the potentiality to act as a cue for retrieval on the DG-CA3 system is also needed in order to sustain memory generalization (even under this modulation). In this regard, it is possible that an absence of contextual cues, like in the NC condition, could prevent a context from acting as a cue for retrieval. Only further experiments will be able to tell us whether even under a higher reduction in the number of cues, DG NMDAR inactivation will still lead to an increase in the level of generalization.

Even though in real life, episodic memory retrieval usually occurs in a degraded context, this process and the balance between pattern completion–pattern separation have not been studied under these conditions. Relatively little work has been done linking remapping and behavior (see Allegra and colleagues [91]). Here, using an incidental memory task, we showed that retrieval of an object in context memory is reflected by the levels of CA3 remapping, demonstrating a clear relationship between remapping and associative episodic-like memory processing. Furthermore, we describe NMDARs as a key player in regulating the balance between retrieval and memory differentiation processes. While an increase in CA3 NMDAR activity boosts memory retrieval, DG NMDAR activity enhances the memory differentiation process. Our results contribute to understanding the adaptive nature of memory, and how it can guide behavior in a way that is consistent with changes in the environmental cues and the internal state of the individual.

## Methods

### Subjects

All the experiments were conducted in accordance with the National Animal Care and Use Committee of the University of Buenos Aires (CICUAL) (CICUAL UF 2021–006 and COPDI-2021-02755266-UBA-DDG-FMED).

For the 2-day version, subjects were 155 male Wistar rats of approximately 250 to 300 g at the beginning of the study. Rats were housed in groups and food deprived to 85% to 90% of their free-feeding weight to increase spontaneous exploration, except during recovery from surgery where food was available ad libitum. Water remained available ad libitum throughout the study.

For the 1-day version experiments, data were obtained from 10 male Wistar rats of approximately 300 to 400 g. All 10 rats were used for the behavioral analysis (Fig 1); 4 rats were implanted for neuronal recordings (Fig 3). Rats were housed separately and were water deprived to 85% to 90% of their free drinking weight to increase spontaneous exploration, except during recovery from surgery, where water was available ad libitum. Food remained available ad libitum throughout the study and water was available for 20 min each day.

For both experimental versions, rats were housed on a reversed 12 h light/12 h dark cycle (lights on 19:00 to 07:00) and all behavioral testing was conducted during the dark phase of the cycle.

### Surgery and cannulation

Rats were bilaterally implanted in the DG, perirhinal cortex (Prh), or CA3 region with 22-gauge indwelling guide cannulas, under ketamine-xylazine anesthesia, as described previously [34,92]. Guide cannulas were implanted according to the following coordinates, measured relative to the skull at bregma Prh (AP −5,5 mm, LL ± 6,6 mm, DV −7,1 mm), DG (AP −3.9 mm, LL ± 1.9 mm, DV −3.0 mm), or CA3 (AP −3.6 mm, LL ± 3.6 mm, DV −3.6 mm). The 1.7 mm lateral separation between DG and CA3 coordinates is enough to distinguish the effects of these infusions, as previous reports have shown that a 0.5 μl volume infusion spreads in a 1.0 mm diameter sphere maintained for up to an hour [93,94].

The cannulas were secured to the skull using dental acrylic and 3 jeweler screws. Obturators, cut to sit flush with the tip of the guide cannulas and with an outer diameter of 0.36 mm, were inserted into the guides and remained there except during infusions. Animals were given at least 7 days to recover prior to drug infusions and behavioral testing.

### Surgery and tetrode implantation

Rats were implanted with a custom-designed 3D-printed microdrive with 8 independently movable tetrodes aiming at CA3, under isoflurane anesthesia, as described previously [95]. Tetrodes were implanted after performing craniotomies in the skull above CA3 (AP: −3.6; ML: ± 4, DV: −3.4). Tetrodes were made of 4 twisted 12 μm tungsten wires (CFW, United States of America). Electrode tips were gold-plated to reduce electrode impedances to around approximately 200 kΩ at 1 kHz and dyed with DiI (Thermo Fisher, USA) to check tetrodes positions. During surgery, tetrode tips were lowered to 1 mm above the structure. After 5 d of recovery, tetrodes were moved gradually until they reached CA3. Neuronal spiking activity and LFP were recorded daily in the home cage and every 2 days in the behavioral task. Units recorded in different sessions were considered independent.

### Infusion procedure

Depending on the experiment, rats received bilateral infusions of Emetine (50 μg μl-1/0.5 μl side; Sigma-Aldrich), D-Cyc (D-4-amino-3-isoxazolidone, 20 μg/μl; Sigma-Aldrich), AP5

(2-amino-5-phosphonopentanoate, 2 µg µl-1/ 0.5 µl side), AMPA/kainate DNQX (6,7-dinitro-quinoxaline-2,3-dione, 1 µg µl-1/0.5 µl side; Sigma-Aldrich), Nimodipine (7.5 µg/0.5 µl, Tocris), or Vehicle (Veh, saline/DMSO 5% for DNQX/DMSO 2%-Tween 2% for ANA-12) at different times during behavioral tasks. Bilateral infusions were conducted simultaneously using two 5-µl Hamilton syringes that were connected to the infusion cannulas by propylene tubing. Syringes were driven by a Harvard Apparatus precision syringe pump, which delivered 0.5 µl to each hemisphere over 2 min. The infusion cannulas were left in place for an additional minute to allow for diffusion. At least 3 days were allowed for washout between repeated infusions. One or 2 days after the behavioral procedure, all animals were infused with 0.5 µl of 5% methyl blue solution through the cannula and were sacrificed 15 min later. Cannula localization was verified and was correct in over 95% of the surgeries. Only the behavioral data for animals with correctly implanted cannulae were included in the analysis.

## Recording procedure

During the recording sessions, neurophysiological signals were acquired continuously at 20 kHz on a 256-channel Amplipex system (http://www.amplipex.com). The microdrive had 3 small leds attached to track the rat's position on the open field that was recorded by a digital video camera at 30 frames/s. LED locations were detected and recorded online with the same software used for the neurophysiological signals.

## Behavioral procedure

The associative retrieval task was performed on a circular open field (90 cm diameter × 45 cm high) made of black plastic surrounded by 6 spatial cues placed above a curtain that enclosed the context preventing visibility of other distal cues in the room. All 6 cues were removable to change their number in different testing conditions. All cues consisted of cardboard and cloth shapes with approximately the same surface ($850 cm^2$). The open field was situated in the middle of a dimly lit room. The open field's floor was covered with wood shavings only for pharmacological experiments. A video camera was placed above the arena and both the sample and choice phase were recorded for later analysis.

For the 2-day version, each rat was handled for 2 days and then habituated to the arena with the corresponding cues for 5 sessions of 10 min/daily before exposure to the objects. In the subsequent training phase, animals were exposed for 10 min to an object in a pseudorandomized position in the arena in the presence of 6 cues, and 24 h after the training phase, a 10-min test phase was performed. During this phase, an identical copy of the object was placed in the same position as during training but in the presence of a variable number of distal contextual cues. This was done to create an object in context memory. In the "all cue" (AC) condition, object presentation occurred in the presence of all the cues presented during training. In the "partial cue" (PC) condition, object presentation was done in a context with half of the cues (3), and in the "two partial cues" (PC2) condition only 2 contextual cues were present. Finally, for the "no cue" (NC) condition, the object was presented in the absence of spatial cues (see Fig 1). For all groups, the training phase was done on AC conditions, in which all 6 distal cues were present.

For the 1-day version of the task, the procedure was similar except that training and test phases were separated by 4 h. In addition, small drops of water were placed in the open field only during habituation to increase context exploration. Animals were habituated to the PC and NC conditions. Moreover, the duration of the training and test phase was 13 min to increase the amount of recorded signal.

For reconsolidation experiments (Figs 2 and 4C), a 3-day protocol was used. First, rats were habituated in both the open field for 5 days and in a triangular arena for 3 days. After

habituation, on day 1, animals were trained by placing an object in the AC context for 10 min as previously; 24 h later (TS1, day 2), a retrieval session was performed. During this session, animals were exposed to an empty context (without any object present) for 10 min with a variable number of contextual cues (depending on the experiment). During a second test session (TS2), 24 h after (day 3), a copy of the original object was presented next to a novel object in the triangular arena for 5 min. For the experiment in Fig 2C, during the first day, animals were trained in the AC context with an object A and in the NC condition with the object B. These 2 training sessions were given in a pseudorandom order and separated by 2 h. During the evaluation, a group was exposed to an empty AC context with all the cues, and another group was exposed to the empty AC context without cues. In all cases, the empty context corresponds to a context without any object present. In a second evaluation, memory for the presented objects (A or B) was tested against 2 different novel objects (C or D) in the triangular arena for 5 min. The triangular arena was made of white foam board and lacked any proximal spatial cues. Each wall was 60 cm long by 60 cm high to prevent the use of distal spatial cues. In this way, for this final day, the memory of 2 familiar objects (experienced originally in different contexts) is tested against 2 novel ones in a novel triangular context. The order of the 2 test sessions was counterbalanced.

For experiments on Figs 2A and 4D, after training on a context-object association as before (day 1), animals were subjected to an AC, PC, or NC contextual re-exposure in the absence of any object, and Emetine or Veh were infused in the Prh immediately after (TS1, day 2). On day 3, the original object memory of day 1 was tested on the triangular context against a novel object (TS2).

## Data analysis

**Experimental design and statistical analysis.** In every experiment, experimental conditions were counterbalanced between trials. For the AR experiments, results were presented as Object exploration %, i.e., percentage of exploration during test phase with respect to training phase ($\frac{t_{test}}{t_{training}} \times 100$). For reconsolidation experiments, results during TS2 session were expressed as a discrimination ratio that was calculated as the time exploring the novel object minus the time exploring the familiar object divided by total exploration time ($\frac{t_{novel} - t_{familiar}}{t_{novel} + t_{familiar}}$). Exploration of an object was defined when the rat had its nose directed towards an object at a distance of 2 cm or less or touching the object with its nose. Leaning on the object with the head facing up does not count as exploration. Climbing or sitting on the object is not included as exploration. For rearing quantification, rearing was defined as the animal climbing into an upright position, resting with hind limbs either in the air or leaning against a wall or object (only when the head was oriented up but not exploring the object). For quantifications, scoring was performed by 2 different experimenters blind to the phase of the task and the experimental condition.

When the animal's position was tracked in the 2-day version of the task, Bonsai (https://bonsai-rx.org/) was used to track position. For the 2-day and 1-day versions, the total distance traveled and the mean instantaneous velocity was calculated for each phase using the animal's position. In addition, for the 1-day version, occupancy maps were constructed with the animal's position for the training and test phase. A Person correlation was computed between training and test phases to control for similar context exploration for the neural analysis.

One sample $t$ tests were used to compare the percentage of exploration with respect to 100%, as a measure of memory. On reconsolidation experiments, one sample $t$ tests were conducted to indicate whether animals have memory (measured as a positive discrimination

ratio) for the objects evaluated. Additionally, paired *t* tests were used to compare between conditions. In the cases where data did not meet normality criteria, Wilcoxon signed-rank test or Mann–Whitney (for unpaired groups) were used instead. For the experiment shown in Fig 1B–1D, the 2-day version of the task, animals were tested 3 times: during the first trial a third of the animals were tested in the AC condition, a third in the PC, and a third in the NC; during the second trial, half of each third was tested in one of the conditions and half in another; and for the third trial, the remaining condition was tested. Animals whose percentage of exploration exceeded 2SD from the group mean were excluded from the analysis. Behavioral response was analyzed using a repeated measures (RM) ANOVA.

For experiments in Fig 1E–1G, the 1-day version of the task, animals were tested every 2 days in one condition of the 3 conditions in a pseudo-randomized order. Behavioral response was analyzed using an RM ANOVA (S1 Fig). Neuronal response in Fig 3C and 3D was analyzed with a Kruskal–Wallis test and in Fig 3F and 3G, a general mixed model was used with rats as a random factor. Data was fitted using a GLM. The best model was selected using the Akaike information criterion (AIC), which takes into account the number of independent variables used to build the model as well as the maximum likelihood estimate (how well the model reproduces the data) and an ANOVA to compare models with increasing complexity. For experiments in Figs 2, 4C, 5B, 5C and 5E, animals were tested twice. During the first trial, half the animals received an infusion of Eme/AP5/D-Cyc (depending on the experiment) and the other half received an infusion of the corresponding Vehicle of each drug. In the second trial, animals received an infusion of Eme/AP5/D-Cyc or the corresponding Vehicle according to what they received in the first trial. For the training phase, the percentage of exploration time for each object was compared using an RM ANOVA. And for the case of Fig 5F, animals were tested 3 times with either Veh/Veh, Veh/Eme, or Nimo/Eme. In Fig 4D, animals were tested in 2 opportunities. In the first trial, half the animals received a Vehicle infusion in the Prh and a Vehicle or AP5 in the CA3/GD, while the other half was infused with Emetine in the Prh and Vehicle or AP5 in the CA3/GD. In the second trial, animals that had received Vehicle in the Prh now had an Emetine infusion and vice versa. Data that did not meet normality criteria was transformed before analysis. The data that support these findings is available in OSF at https://osf.io/7pw23/.

**Putative internal representation of the context.** To study if our behavioral data fitted a dichotomic distribution, we performed a bootstrap LRT for assessing the number of mixture Gaussian components that could model the behavioral data. We used an equal variance model with over 900 replications.

To separate sessions of the 1-day version of the task in terms of the animal's dichotomic behavioral output (sessions where animals discriminated between contexts from sessions where they did not), we used the distribution of the behavioral output variable (percentage of object exploration) from the 2 session groups that showed memory retrieval (AC and PC). Sessions with an object exploration rate < 2-SD of the mean were considered as sessions in which animals recognized the context as the same one experienced in training (retrieval sessions). Sessions with an object exploration rate > 2-SD were classified as sessions where rats behaved as if the context was different from the one experienced in training (non-retrieval). For the analysis of the categorical data in the 1-day version, we performed a general mixed model with rats as a random factor. Similarly, for the 2-day version of the task, we used the pooled distribution of the AC and PC object exploration values of the independent experiment of Fig 1D to determine the +2SD threshold for memory retrieval. When object exploration values were above this threshold from the pooled mean, animals were considered as not recognizing the context as the same as the training (non-retrieval). For Figs 4–6, object exploration values that exceed this 2SD threshold were represented as red dots in the graphs. To analyze the

categorical data in the 2-day version, we performed MacNemar's Chi-square tests considering whether animals performed above or below 2SD in the Veh condition and whether they changed their performance with the different drug infusions. Experiments were pooled according to the effect of each drug. The "memory boosting" group includes the experiments of D-Cyc in CA3 and AP5 on the DG (both on the PC2 condition) and the "memory impairing" group includes the experiments of the D-Cyc on the DG and the AP5 on CA3 (both in the PC condition).

**Internal representation controls.** To rule out nonspecific effects of clustering by internal representation, 2 controls were performed. First, the spatial correlation and firing rate change analyses were repeated using the activity of the recorded non-place cells. We randomly selected an equal number group of non-place cells (S7D Fig). On the other hand, instead of using object exploration to define the internal representation groups, the time spent in an area similar in size to and opposite to the object was used to calculate the percentage of time spent in the area in the test relative to training ($\frac{t_{test}}{t_{training}} \times 100$). The percentage of exploration of the control area was used to establish a new threshold for internal representation. Place cells were sorted according to this new criterion and the spatial correlation and firing rate analysis were repeated (S7E Fig).

**Spike sorting and place cells.** Neurophysiological and behavioral data were explored using NeuroScope (http://neurosuite.sourceforge.net) [96]. Spikes were sorted in 2 steps, first automatically, using custom-made software, KlustaKwik (http://klustakwik.sourceforge.net) [97], followed by manual adjustment of the clusters (using Klusters software package; http://klusters.sourceforge.net) [96]. Only units with clear refractory periods and well-defined cluster boundaries were taken into account for the analysis (Harris and colleagues [97]). To be included in the analysis, only units with a stable recording during the whole session were considered. Briefly, animals were recorded in their home cage for 10 min after the training phase and after the test phase until they fell asleep. If a unit was not active in one of the visits to the open field, it must be active in the home cage to be included in the analysis. We recorded a total of 1,307 well-isolated units from CA3 of 4 freely moving rats in 24 sessions. Of these, 332 were considered putative place cells. Units were considered place cells if they fulfilled 3 criteria: (a) their spatial information content [98] was higher than 0,15 bit/spike; (b) the information content of the unit was different from chance level (computed by shuffling the spike train of the cell and rat position 100 times, calculating the information content each time, and establishing the quantile 99% as threshold) in at least one of the phases of the recording day; and (c) their mean firing rate was higher than 0.1 Hz in at least one of the phases of the recording day.

**Place cell activity.** Only data recorded during epochs when the rat was moving faster than 10 cm/s were used (exploratory epochs). Spiking data and animal's position were sorted into 3-cm × 3-cm bins to create raw maps of spike counts and occupancy. A Gaussian kernel filter (s.d. = 9 cm) was applied to both maps. A smoothed rate map was constructed by dividing the smoothed spike map by the smoothed occupancy map. The smoothed rate maps of the training and test phase were used to compute the mean and peak firing rates in the open field as well as the spatial correlation and firing rate remapping.

**Spatial correlation and firing rate change.** In this analysis, bins visited for less than 15 msec and less than 4 times were excluded to avoid artefacts. Only bins that meet these requirements in both test and training phases were considered. Moreover, only sessions with 5 neurons or more were included in the analysis. For the spatial correlation measure, smooth rate maps of individual place cells in the training and test phase were compared with a bin-by-bin Pearson's correlation. Firing rate change between the training phase and the test phase was estimated for each cell as the absolute value of dividing the difference between the mean firing rate of the training and testing phase, and their sum ($|\frac{Fr_{training} - Fr_{test}}{Fr_{training} + Fr_{test}}|$).

To control for the effect of different sampling patterns between the training and test phases on these parameters, subsampled firing maps were constructed and used to compute the spatial correlation and the firing rate change [99]. Spike and position data in each spatial bin were subsampled to match the minimum number of samples in the corresponding bin of the less visited bin of the training or test. Examples of the subsampled firing maps are shown in S5B Fig.

**Population vector and t-SNE analysis.** For the population analysis, we use sessions with more than 6 neurons. The same bin exclusion criteria described in the previous section were applied. In the populational vector analysis, for each valid session, rate vectors were constructed by arranging the firing map of each neuron for each phase (training and test) in an x-y-z stack, where x and y represent the 2 spatial dimensions and z represents the cell identity index [22]. The distribution of mean firing rates in the z-axis for a given x-y location represents the population vector for that location bin. A bin-by-bin correlation coefficient was calculated between session's phases and the accumulated frequency was plotted.

## Histology

After behavioral testing, rats were anesthetized by IP injection with 2 ml of Euthatal (Rhône Merieux) and perfused transcardially with phosphate-buffered saline (PBS), followed by 10% neutral buffered formalin. The brains were removed and postfixed in formalin for at least 24 h before being immersed in 20% sucrose solution until they sank. To verify the site of the infusion, 60-μm sections of the brain were cut on a freezing microtome encompassing the extent of the injector track. Every fifth section was mounted on a gelatin-coated glass slide and stained with cresyl violet. Slides were examined under a light microscope to verify the location of the injections (S2A–S2D and S2G Fig). To verify the tetrode's position, 70-μm sections of the brain were cut on a freezing microtome. Each section of a relevant part of the hippocampus was mounted on a gelatin-coated glass slide and was examined under a red fluorescent microscope. All tetrodes of the 8-tetrode bundle were identified by finding the tip of each electrode across sections (S2E and S2F Fig).

## Supporting information

**S1 Fig. (A) and (B)** Repeated measure analysis. **(A)** Total object exploration during the training session for AC, PC, and NC in the different expositions to the same condition during the 1-day version of the task. Two-way RM ANOVA, $n = 10$, sessions = 75, each dot represents a session, interaction: F = 0.5755, $p = 0.7484$. AC: 47.89 ± 3.38, PC: 47.39 ± 3.07, NC: 49.74 ± 3.50. **(B)** Total object exploration percentage for AC, PC, and NC in the different expositions to the same condition during the 1-day version of the task. Two-way RM ANOVA, $n = 10$, sessions = 75, each dot represents a session, F = 16.04, $p < 0.0001$. Tukey's post hoc test: **** $p < 0.0001$ AC vs. NC; $p = 0.25$ AC vs. PC; ** $p = 0.001$ NC vs. PC. One sample $t$ test against 100% AC t = 14.45, $p < 0.0001$; PC t = 8.76, $p < 0.0001$; NC t = 0.37, $p = 0.91$. AC: 56.98 ± 3.31, PC: 63.93 ± 4.11, NC: 92.13 ± 7.34. The re-exposures to the different conditions have no effect on the exploration time in the training session or in the object exploration percentage. **(C)** Percentage of rearings during the test phase in relation to training for the AC, PC, and NC conditions in the 2-day version. One-way RM ANOVA F = 4.26, $p = 0.048$, AC: 93.57 ± 13.18, PC: 101.10 ± 8.28, SC: 132.9 ± 12.53. $n = 11$. * $p < 0.1$. While no previous differences between groups in rearings were present during the training session (RM one-way ANOVA, F = 0.30, $p = 0.69$), post hoc comparisons revealed a marginally significant increase in the NC condition when compared to both the AC and PC conditions (AC vs. NC: t = 2.23, $p = 0.05$, NC vs. PC: t = 2.14, $p = 0.06$). To note, unlike object-related exploration, contextual

exploration in the AC condition (as manifested by the number of rearings) does not decrease between training and test since animals were habituated to the AC context (and not to the object) prior to training session (one sample $t$ test from 100%, AC t = 0.49, $p$ = 0.636, PC t = 0.13, $p$ = 0.896, NC t = 2.62, $p$ = 0.025). Individual values used to calculate the mean and SEM are presented as dots. The data that support these findings is available in OSF at https://osf.io/7pw23/.
(TIF)

**S2 Fig.** **(A)** Safranin staining image of a coronal brain section of a rat cannulated in the CA3 of HP. **(B)** Schematic representation of coronal sections of the rat brain with representative infusion area in an example experiment. **(C)** Safranin staining image of a coronal brain section of a rat cannulated in the CA3 of HP. **(D)** Schematic representation of coronal sections of the rat brain with representative infusion area in an example experiment. **(E)** Two representative coronal brain sections with tetrode trajectory. **(F)** Schematic representation of the tetrode recording sites for 4 recorded rats in CA3. Each cross represents a tetrode tip. See full methods for verification of tetrode recording sites. **(G)** Safranin staining image of a coronal brain section of a rat cannulated in the PRH. Scale: 200 μm and schematic representation of coronal sections of the rat brain with representative infusion area in an example experiment. All schematic representations of brain slices are adapted from Swanson's Brain map: structure of the rat brain (Swanson, 2004), levels 32 and 37, which is distributed under the terms of a Creative Commons Attribution-Noncommercial 4.0 license CC BY-NC-4.0 (https://creativecommons.org/licenses/by-nc/4.0/). As such it is not covered by the CC BY 4.0 license and further reproduction of these panels would need to follow the terms of the CC BY-NC 4.0 license. The data that support these findings is available in OSF at https://osf.io/7pw23/.
(TIF)

**S3 Fig.** **(A)** Percentage of total distance traveled during test on Fig 5F. No significant difference was found in the distance traveled by animals after infusion of DCyc in CA3 compared to Veh. Paired $t$ test, $p$ = 0.949, t = 0.06. Veh: 93.11 ± 12.16, DCyc: 93.90 ± 6.18. **(B)** Percentage of average speed in cm per second of animals injected in CA3 with DCyc compared with Veh. Paired $t$ test, $p$ = 0.548, t = 0.63. Veh: 97.10 ± 12.60, DCyc: 108.00 ± 12.26. **(C)** Percentage of total distance traveled during test over training in the experiment in Fig 5E. No significant differences were found between the distance traveled by animals after infusion of DCyc in the DG compared to Veh. Paired $t$ test, $p$ = 0.877, t = 0.16. Veh: 107.7 ± 23.60, DCyc: 103.3 ± 13.81. **(D)** Percentage of average velocity in cm per second during the test over training in Fig 5E. No significant difference was found in the speed of animals injected in the DG with DCyc compared to Veh. Paired $t$ test, $p$ = 0.841, t = 0.21. Veh: 120.00 ± 27.17, DCyc:115.2 ± 9.71. The data that support these findings is available in OSF at https://osf.io/7pw23/.
(TIF)

**S4 Fig.** **(A)** Percentage of total distance traveled during test over training in the experiment in Fig 3. No significant differences were found between the distance traveled in each condition, $n$ = 24, one-way ANOVA, $p$ = 0.62. Tukey's post hoc test: $p$ = 0.99 AC vs. NC; $p$ = 0.33 AC vs. PC; $p$ = 0.37 NC vs. PC. AC: 92.26 ± 3.29, PC: 100.8 ± 3.81, NC: 92.89 ± 3.11. **(B)** Percentage of average velocity in cm per second during the test over training in Fig 3. No significant difference was found in the speed of animals between conditions, $n$ = 24, one-way ANOVA, $p$ = 0.93. Tukey's post hoc test: $p$ = 0.98 AC vs. NC; $p$ = 0.93 AC vs. PC; $p$ = 0.97 NC vs. PC. AC: 93.91 ± 0.03, PC: 92.17 ± 0.04, NC: 94.07 ± 0.03. **(C)** Pearson correlation between the occupancy maps of the training and test phase. No significant difference was found in the occupancy maps of the training and test phases between conditions, $n$ = 24, one-way ANOVA,

$p = 0.91$. Tukey's post hoc test: $p = 0.94$ AC vs. NC; $p = 0.99$ AC vs. PC; $p = 0.93$ NC vs. PC. AC: $0.66 \pm 0.06$, PC: $0.64 \pm 0.06$, NC: $0.64 \pm 0.04$. The data that support these findings is available in OSF at https://osf.io/7pw23/.
(TIF)

**S5 Fig.** **(A)** Normalized firing maps of Fig 3. Place maps for training session (left) and test session (right) for each condition. Extreme's firing maps were normalized by the max firing rate of each neuron and the middle firing map was normalized by the max firing rate of the phase with higher firing rate peak. Spatial correlation and firing rate change were calculated for each neuron. **(B)** Original and subsampled firing maps. Subsampled firing maps were constructed by down-sampling the spike and position data in each spatial bin to match the minimum number of samples in the less visited bin of the training or test. The data that support these findings is available in OSF at https://osf.io/7pw23/.
(TIF)

**S6 Fig.** **(A)** Percentage of place cells over the total number of recorded neurons in each session. No significant differences were found between conditions. $n = 24$, Kruskal–Wallis test $p = 0.99$ AC vs. NC; $p = 0.86$, AC vs. PC; $p = 0.88$ NC vs. PC. AC: $25.65 \pm 3.21$, PC: $21.59 \pm 3.84$, NC: $25.22 \pm 4.86$. Context enrichment does not increase the proportions of place cells coding a context. **(B)** Difference in spatial information index between test and training. No change in spatial coding between conditions. $n = 54–141$, Kruskal–Wallis test NC-AC $p = 0.93$, NC-PC $p = 0.88$, and AC-PC $p = 0.98$. AC: $0.01 \pm 0.03$, PC: $0 \pm 0.03$, NC: $0.02 \pm 0.03$. **(C)** Difference in place field size between test and training. No significant differences in place field size between conditions. $n = 54–141$, Kruskal–Wallis test NC-AC $p = 0.96$, NC-PC $p = 0.78$, and AC-PC $p = 0.67$. AC: $-5.21 \pm 3.43$, PC: $-2.43 \pm 6.58$, NC: $-5.59 \pm 4.37$. **(D) and (E)** Correlation between place cell activity and memory output. **Spatial correlation:** Pearson's correlation between spatial correlation and object exploration %, $n = 332$, $p = 1.47\text{e-}10$, R = $-0.34$. **Firing rate change:** Pearson's correlation between firing rate change and object exploration %, $n = 332$, $p = 0.00015$, R = $0.20$. The data that support these findings is available in OSF at https://osf.io/7pw23/.
(TIF)

**S7 Fig.** **(A)** Object exploration percentage histogram. Gray line represents the normal distribution of the = sessions, red line represents the normal distribution of the ≠ sessions. **(B) Spatial correlation** (left) and **firing rate change** (right) sorted by internal representation using only place cells activity from NC condition, $n = 65–72$, LMM, p spatial correlation = $0.0007$, =: $0.52 \pm 0.03$, ≠:$0.25 \pm 0.04$; p firing rate change = $0.1742$, =: $0.22 \pm 0.02$, ≠:$0.36 \pm 0.04$. **(C)** Population vector. The population vector analysis follows the same tendency that the single unit analysis. **(D)** Non-place cells have not a role encoding animal internal representation of context. A number of neurons equal to the one in Fig 3 was selected randomly from the subset of recorded neurons that did not fulfill place cell criteria and an analysis equivalent to the one in Fig 3 was applied. Right: No place cell **spatial correlation** sorted by internal representation. $n = 65–267$, LMM, $p = 0.48$, =: $0.34 \pm 0.01$, ≠:$0.37 \pm 0.03$; left: no place cell **firing rate change** sorted by internal representation. $n = 65–267$, LMM, $p = 0.37$, =: $0.10 \pm 0.01$, ≠:$0.08 \pm 0.02$. **(E)** Place cells activity sorted by a threshold determined by the exploration of an area opposite the object. Right: **spatial correlation** sorted by control internal representation. $n = 65–267$, LMM, $p = 0.87$, =: $0.46 \pm 0.02$, ≠:$0.54 \pm 0.03$. Left: **firing rate change** sorted by control internal representation. $n = 65–267$, LMM, $p = 0.78$, =: $0.26 \pm 0.02$, ≠:$0.19 \pm 0.02$. The data that support these findings is available in OSF at https://osf.io/7pw23/.
(TIF)

## Acknowledgments

We thank Bárbara Giugovaz Tropper and Victoria Deschamps for her technical assistance, David Jaime for his help with animal care, and Dr. Jorge Medina for helpful discussions and sharing of some of the drugs. We thank Noelia Weisstaub and Joaquin Piriz for their comments on the manuscript.

## Author Contributions

**Conceptualization:** Magdalena Miranda, Azul Silva, Mariano Belluscio, Pedro Bekinschtein.

**Data curation:** Magdalena Miranda, Azul Silva.

**Formal analysis:** Magdalena Miranda, Azul Silva.

**Investigation:** Magdalena Miranda, Azul Silva, Juan Facundo Morici, Marcos Antonio Coletti.

**Supervision:** Mariano Belluscio, Pedro Bekinschtein.

**Writing – original draft:** Magdalena Miranda, Azul Silva, Mariano Belluscio, Pedro Bekinschtein.

**Writing – review & editing:** Magdalena Miranda, Azul Silva, Mariano Belluscio, Pedro Bekinschtein.

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
