## [Editor Report · Decision Letter 0]

13 Oct 2023

Dear Dr Bekinschtein, 

Thank you for submitting your manuscript entitled "Behavioral, molecular and neuronal mechanisms involved in spatial recognition memory retrieval in the rat hippocampus" for consideration as a Research Article by PLOS Biology.

Your manuscript has now been evaluated by the PLOS Biology editorial staff as well as by an academic editor with relevant expertise and I am writing to let you know that we would like to send your submission out for external peer review.

Once your full submission is complete, your paper will undergo a series of checks in preparation for peer review. After your manuscript has passed the checks it will be sent out for review. To provide the metadata for your submission, please Login to Editorial Manager (https://www.editorialmanager.com/pbiology) within two working days, i.e. by Oct 15 2023 11:59PM.

Kind regards,

Christian

Christian Schnell, PhD

Senior Editor

PLOS Biology

cschnell@plos.org

---

## [Decision Letter · Decision Letter 1]

20 Nov 2023

Dear Dr Bekinschtein,

Thank you for your patience while your manuscript "Behavioral, molecular and neuronal mechanisms involved in spatial recognition memory retrieval in the rat hippocampus" was peer-reviewed at PLOS Biology. It has now been evaluated by the PLOS Biology editors, an Academic Editor with relevant expertise, and by several independent reviewers. 

You will find their reports at the end of this email.

As you will see below, Reviewers 1 and 3 are quite positive about your study, while Reviewer 2 questions the conceptual advance. After feedback from the reviewers on each other's reports and discussing the reports and the additional feedback with the Academic Editor, we would like to invite you to revise the work to thoroughly address the reviewers' reports. We don't think that new experimental evidence is required to address the reviewers' concerns but encourage you to provide the requested additional analyses and textual revisions by all reviewers. One specific suggestion that came up is to control for the specific pattern of sampling between the TR and TS sessions by down-sampling the data to match the minimum number of samples at each visited location when comparing rate maps, as explained in (Keinath et al., 2018, eLife), to ensure that you’re not simply reading out the behavioral biases in location.

Given the extent of revision needed, we cannot make a decision about publication until we have seen the revised manuscript and your response to the reviewers' comments. Your revised manuscript is likely to be sent for further evaluation by all or a subset of the reviewers.

**IMPORTANT - SUBMITTING YOUR REVISION**

*Re-submission Checklist*

*Published Peer Review*

*PLOS Data Policy*

*Blot and Gel Data Policy*

Sincerely,

Christian

Christian Schnell, PhD

Senior Editor

PLOS Biology

cschnell@plos.org

REVIEWS:

Reviewer #1: Understanding whether and how contextual representations guide memory retrieval is a fundamental question in neuroscience which must be addressed at multiple levels of explanation - from synapses to circuits. Here, Miranda et al. add to our knowledge of this process by presenting various pieces of causal and correlational evidence linking protein synthesis, NMDA receptor activation, single-unit activity, and behavioral assays of context-object memory in rats.

The workhorse of this study is a paired object-context learning paradigm, with object exploration times serving as the dependent of object memory, and manipulations to the cues which comprise the context. When exposed to just a subset of visual cues previously present during a context-object association (partial cue - PC), behavioral evidence indicates that an internal representation of context is retrieved despite degraded sensory cues. Interfering with protein synthesis in perirhinal cortex following a second exposure to the context of a context-object association disrupts the memory for the paired object, but does not disrupt the memory for objects encountered in other environments. Electrophysiological recording of place cells in CA3 indicates that the hippocampal map remains stable even in the degraded PC condition, and that the amount of remapping in CA3 in an extra-degraded no-cues condition correlates with the behavioral evidence of a change in the internal representation of context. 

Next, the authors investigate the possible role of NMDARs in mediating context-object memory. Direct infusion of an NMDAR antagonist in CA3 prior to a second experience of a context-object pair in a PC condition led to rats treating the object as if it were a new context-object pair; this was not the case in the all-cues condition, suggesting that NMDARs in CA3 play a necessary role in reinstatement of degraded memories but is not necessary for the reinstatement of fully-cued memories. Combining both the CA3-NMDAR-Antagonist prior to a second contextual exposure with perirhinal protein synthesis inhibition following a second contextual exposure in either the fully-cued or PC context yielded corroborating evidence of this selective contribution of CA3 in degraded environments. Turning to the DG, the authors show in various PC conditions that activating or antagonizing NMDARs in the DG leads to behavioral evidence of memory discrimination or generalization, respectively. Bringing this full circle, the authors then show that in a PC condition activating NMDARs in CA3 yields behavioral evidence of memory generalization which depends on L-type voltage gated calcium channels, effectively demonstrating bidirectional modulation of opposing discrimination/generalization dynamics in CA3 and DG.

Altogether, I find this to be very good work. There writing and narrative are clear, there is an impressive amount of data without becoming overwhelming, and the conclusions are sensible. I have only minor suggestions which could help to improve the paper but should not be prohibitive to publication.

Minor concerns:

For the CA3 electrophysiology results, I would consider including analysis of population vector similarity to complement your findings. A population vector measure summarizes the similarity of the neural population as a whole recorded in a session, and would allow you to perform an animal-level analysis rather than treating each place cell as independent. This might clarify some of the variability you observe across cells. 

For your internal representation GLM - the model with more variables fits best, but perhaps needs to be controlled for the number of variables (2 > 1). Maybe compared to a shuffled version of the variable of interest, and/or report the marginal gain in R2?

For the Figure 4 caption and related text, the acronyms TC and CP are used. I'm not sure if I'm missing their reference or if they are referring to AC and PC conditions, respectively (on my reading, I assumed the latter). This might be me making a mistake but I would double-check those referents and/or clarify their meaning as they occur.

Reviewer #2: In this study, the authors exposed rodents to differently enriched environments and tested their exploration time of a familiar object in relation to a novel object, which animals have never seen before. They find that in dependence on the number of external cues surrounding the environment the recall of a context and the differentiation between the familiar and novel object is higher in enriched contexts (ACs) than in poor in 'non cued' (NC) environments. Next, they block protein synthesis by injecting Emetine in the perirhinal cortex and find that the discrimination between the familiar and novel object is only reduced in the cue-enriched AC context but not the non-cued NC context. The authors conclude first, that protein synthesis during consolidation is needed for the recall of object-context memory; and (2) that contexts are capable of guiding reactivation of 'relevant' object memory, while memories of 'non-relevant objects' are not susceptible to the action of protein synthesis inhibitors (page 4, second paragraph). This conclusion is an over-interpretation of the data and actually not supported by the data. What is the definition on a 'relevant' and 'not-relevant memory'? What seem to matter is the order of context presentations. If a cue-enriched context is presented a second time after the initial object-context pairing than the animal memorizes in the test phase the familiar object and explores the novel one. The second exposure to the enriched context seems to be important to support the association in the object-context memory. If the animal is exposed to an empty arena (NC context) than the association between object and the initial context seemed to be markedly reduced. However, this has nothing to do with relevant or on-relevant object memory. Why was protein synthesis blocked in the perirhinal cortex and not e.g. in the dentate gyrus well known to store information for a live time (see below)? Next, the authors perform single unit recordings within CA3 during exploration of enriched contexts (TR 6 cues vs TR, 6 cues; TR 6 cues versus TR 3 cues; TR 6 cues versus no cue), and find relationships between the three different context comparisons and spatial parameters such as spatial tuning and exploration time of the object. What is missing is the number of place cells and place field correlations. Finally, the authors apply AP5 infusion to block NMDA receptors in the dentate gyrus or the CA3 area and show that this interference causes the reduced animals' ability to discriminate between objects (pattern separation) a well-known phenomenon previously observed by McHugh et al (Nature 2007). In summary, most of the findings relate to previously published findings, which lowered my enthusiasm for the study. On several occasions, the authors relate their work to episodic memory but the work does not reflect episode of information at all. Statements such as in the abstract last sentence: 'They also provide new insights into molecular mechanisms underlying the communication between hippocampal subregions', is overestimated because this study does not provide any molecular mechanisms explain inter-hippocampal communication. There are several similar locations within the text, which require a thorough revision of the text. I final criticism is the embedding of the single unit recording data. They seem to be much unrelated to the rest of the manuscript. 

Major comments:

1. The authors argue that exposure of the object within an enriched (AC) context requires protein synthesis in the perirhinal cortex (PrC) because infusion of the PrC with Emetine reduces object discrimination if the familiarized object is combined in the AC context with a novel object. Thus, this is an object recognition task discrimination is based on the recognition of the novel previously unexperienced object. The less cues are available in the environment the more seems the animal to focus on the object. In a cue-lacking environment the animals has nothing else to do than to explore the only object in the arena present. Thus, it is not necessarily the object-context relationship, which drives the stronger observation of the object but the pure lack of any other visual or tactile inputs. 

2. The triangles in Figure 2 are confusing, because triangular environments are shown with the label AC implying that animals are exposed to enriched environments. Thus, external cues should be present in this part of the figure. 

3. It remain unclear why Emetine was injected in the perirhinal cortex because recent studies showed that context memory is encoded and stored within the dentate gyrus for several days up to weeks and infant memories can be recalled upon reactivation of principal cell assemblies in the adult rodent (Kim et al., Nature Comm 2020; Guskjolen et al., Curr Biol, 2018). Thus, the dentate gyrus seem to keep contextual information for long periods of time. What would have happened on the behavioral level if Emetine was injected in the DG or in a different brain area not involved in the binding of objects and context?

4. One of the main conclusions of Figure 2 is that the context is able to guide reactivation of 'relevant' object memories, but what is here a 'relevant' object memory? Indeed the data shown in Figure 2 indicate that the second recall of the enriched context is important to make an association between the object and the context and allows the identification of the novel object. It has been previously shown in object recognition tasks that 24 h after initial environmental exposure memory is stored and can be recalled in object recognition tasks. It is also very clear from the literature that distal cues within the environment are important to identify a context and help the animal to memorize a given context. Thus, exposure of rodents to a NC condition with lacking cues makes a 'remembering' or consolidation of a memory quite difficult. Therefore some of the here presented results are not surprising. 

5. Another finding of the study

---

## [Decision Letter · Decision Letter 2]

24 Apr 2024

Dear Dr Bekinschtein,

Thank you for your patience while we considered your revised manuscript "Behavioral, molecular and neuronal mechanisms involved in spatial recognition memory retrieval in the rat hippocampus" for publication as a Research Article at PLOS Biology. This revised version of your manuscript has been evaluated by the PLOS Biology editors, the Academic Editor and two of the original reviewers.

Based on the reviews and on our Academic Editor's assessment of your revision, we are likely to accept this manuscript for publication, provided you satisfactorily address the following data and other policy-related requests.

* We would like to suggest a different title to improve readability: "Retrieval of contextual memory can be predicted by CA3 remapping and is differentially influenced by NMDAR activity in rat hippocampus subregions"

* Please add the links to the funding agencies in the Financial Disclosure statement in the manuscript details.

* Please note that per journal policy, the model species studied should be clearly stated in the abstract of your manuscript. 

* Please note that per journal policy, we do not allow the mention of "data not shown", "personal communication", "manuscript in preparation" or other references to data that is not publicly available or contained within this manuscript. Please either remove mention of these data or provide figures presenting the results and the data underlying the figure(s).

DATA POLICY:

Regardless of the method selected, please ensure that you provide the individual numerical values that underlie the summary data displayed in the following figure panels as they are essential for readers to assess your analysis and to reproduce it: 1CDFG, 2BDE, 3BCDEF, 4CD, 5BCEFG, 6A, S1ABC and S3E

We expect to receive your revised manuscript within two weeks. 

*Published Peer Review History*

*Press*

Sincerely,

Christian

Christian Schnell, PhD

Senior Editor

cschnell@plos.org

PLOS Biology

Reviewer remarks:

Reviewer #1 (Alexandra Keinath): After reading through the author's responses to my concerns and the concerns of other reviewers, I feel that these were satisfactorily addressed.

Reviewer #3: I am satisfied with the changes the authors have made to their manuscript, they have addressed my main concerns and I believe the manuscript's findings are stronger now.

---

## [Editor Report · Decision Letter 3]

24 May 2024

Dear Dr Bekinschtein,

Thank you for your patience while we considered your revised manuscript "Retrieval of contextual memory can be predicted by CA3 remapping and is differentially influenced by NMDAR activity in rat hippocampus subregions" for publication as a Research Article at PLOS Biology. 

Thank you also for addressing most of the editorial requests. A few points, however, are still open:

* Thank you for depositing the source data at OSF. The repository is not accessible though, so we cannot see the data that have been deposited. Can you please make the repository publicly available?

* Please note that per journal policy, we do not allow the mention of "data not shown" or other references to data that is not publicly available or contained within this manuscript. Please either remove mention of these data or provide figures presenting the results and the data underlying the figure(s). There is currently one instance of this in the "Surgery and tetrode implantation" section. 

* Finally, please remove the information to the funders from the Acknowledgement section and provide this in the manuscript details directly in Editorial Manager. Please add there also the links to the funders. 

We expect to receive your revised manuscript within two weeks. 

*Published Peer Review History*

*Press*

Sincerely,

Christian

Christian Schnell, PhD

Senior Editor

cschnell@plos.org

PLOS Biology

---

## [Editor Report · Decision Letter 4]

12 Jun 2024

Dear Pedro,

Thank you for the submission of your revised Research Article "Retrieval of contextual memory can be predicted by CA3 remapping and is differentially influenced by NMDAR activity in rat hippocampus subregions" for publication in PLOS Biology. On behalf of my colleagues and the Academic Editor, Jozsef Csicsvari, I am pleased to say that we can in principle accept your manuscript for publication, provided you address any remaining formatting and reporting issues. These will be detailed in an email you should receive within 2-3 business days from our colleagues in the journal operations team; no action is required from you until then. Please note that we will not be able to formally accept your manuscript and schedule it for publication until you have completed any requested changes.

PRESS

Sincerely, 

Christian

Christian Schnell, PhD

Senior Editor

PLOS Biology

cschnell@plos.org